# Local Mechanisms of Compositional Generalization in Conditional Diffusion

**Arwen Bradley** [1]

## Abstract

Conditional diffusion models appear capable of compositional generalization, i.e., generating convincing samples for out-of-distribution combinations of conditioners, but the mechanisms underlying this ability remain unclear. To make this concrete, we study length generalization, the ability to generate images with more objects than seen during training. In a controlled CLEVR setting (Johnson et al., 2017), we find that length generalization is achievable in some cases but not others, suggesting that models only sometimes learn the underlying compositional structure. We then investigate locality as a structural mechanism for compositional generalization. Prior works proposed score locality as a mechanism for creativity in unconditional diffusion models (Kamb & Ganguli, 2024; Niedoba et al., 2024), but did not address flexible conditioning or compositional generalization. In this paper, we prove an exact equivalence between a specific compositional structure (*conditional projective composition*) (Bradley et al., 2025) and scores with sparse dependencies on both pixels and conditioners (*local conditional scores*). This theory also extends to compositions of concepts (such as style+content) in feature-space. We validate our theory empirically: CLEVR models that succeed at length generalization exhibit local conditional scores, while those that fail do not. Furthermore, we show that a causal intervention explicitly enforcing local conditional scores enables length generalization in a previously failing model. Finally, we investigate SDXL and find that in pixel-space, spatial locality is present but conditional-locality is mostly absent; however, we find quantitative evidence of local conditional scores in the network's learned feature-space.

[1] Apple, Cupertino CA. Correspondence to: Arwen Bradley <arwen_bradley@apple.com>.

*Proceedings of the 43rd International Conference on Machine Learning*, Seoul, South Korea. PMLR 306, 2026. Copyright 2026 by the author(s).

## 1. Introduction

Conditional diffusion models (Sohl-Dickstein et al., 2015; Ho et al., 2020; Song & Ermon, 2019; Song et al., 2020) appear to possess remarkable compositional generalization capabilities. For example, text-to-image models (Dhariwal & Nichol, 2021; Rombach et al., 2022; Ramesh et al., 2022) generate convincing images for prompts like "a photograph of a cat eating sushi with chopsticks" that were (probably) not seen during training. These models may generalize by composing known concepts (e.g. cat+sushi) in novel ways. However, the extent of generalization in large-scale models is unclear as their train sets are not publicly known (perhaps they *have* seen cats eating sushi). Further, despite recent progress (Okawa et al., 2024; Park et al., 2024; Sclocchi et al., 2025; Kadkhodaie et al., 2023; Favero et al., 2025; Chen et al., 2024; Wang et al., 2024; Lukoianov et al., 2025), the mechanisms underlying compositional generalization remain unclear.

We first propose a concrete and controlled setting in which to study compositional generalization: length generalization in location-conditioned models trained on CLEVR (Johnson et al., 2017), a synthetic dataset of objects with various locations, shapes, and colors. Length generalization refers to the ability to generate more objects than seen in training – e.g., can a location-conditioned model trained on 1-3 objects and tested on $K > 3$ locations actually generate images with $K$ objects at the correct locations? Prior work demonstrated length generalization of *explicit* composition of multiple diffusion models via linear score combination (Du et al., 2023; Liu et al., 2022; Bradley et al., 2025). In contrast, we study length generalization of a *single* conditional model. By training on multi-object samples, we hope that this model can learn the underlying compositional structure of the data, hence length-generalize. We find empirically that, depending on conditioning and architecture specifics, these models sometimes length-generalize and sometimes do not.

Next, we study local mechanisms for compositional generalization. We build primarily on two lines of prior work: one on local mechanisms for creativity, and another on compositionality in diffusion. First, (Kamb & Ganguli, 2024; Niedoba et al., 2024) recently proposed that models learn *local score functions*, enabling creativity via mosaicing of

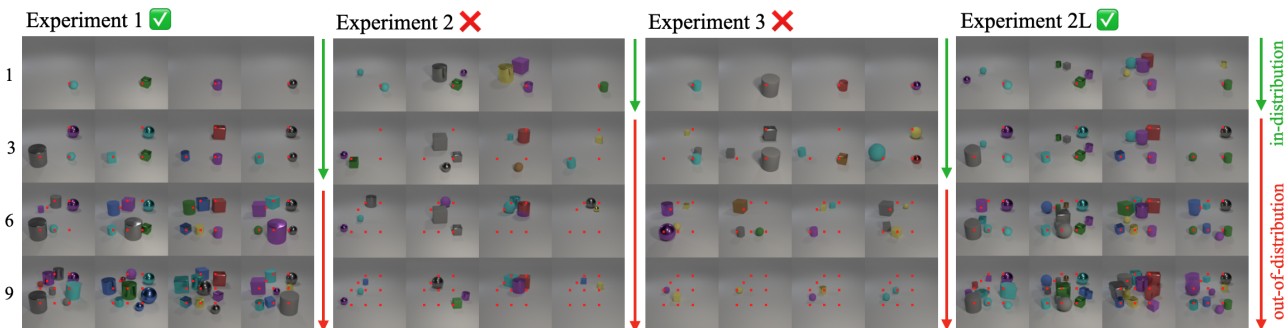

*Figure 1.* **Length generalization in location-conditioned CLEVR models.** We study length generalization in location-conditioned models trained on images with 1-3 objects, and tested on 1, 3, 6, 9 locations (6, 9 are OOD), with red dots indicating the conditioned locations at test-time. For each experiment, the rows correspond to different conditioners (1, 3, 6, or 9 locations) and the columns show 4 different samples. All models have the same architectures and training data and differ only in the design of their conditioners (see Figure 7). In **Experiment 1**, a grid-style conditioner labels the locations of all objects in the scene; the model successfully length-generalizes up to 9 locations. In **Experiment 2**, a grid-style conditioner labels the location of only a single object (randomly selected); the model fails to length-generalize (in this case, even 3 locations is OOD). In **Experiment 3**, a list-style conditioner labels the locations of all objects; this model fails to length-generalize beyond 3 objects. Additional samples shown in Figure 8. **Experiment 2L** applies a causal intervention to the failing Exp. 2: we modify the model architecture to explicitly enforce local conditional scores, use the same training data and conditioning as Exp. 2, and find that Exp. 2L length-generalizes while Exp. 2 failed (see also Section E.4 for an analogous Exp. 3L).

local patches from different images. These works only study unconditional and class-conditional diffusion, however, and do not consider flexible conditioners such as those used in text-to-image diffusion, which are central to questions of compositional generalization. It therefore remains unclear whether local mechanisms are relevant to compositional generalization in conditional diffusion models. Second, (Bradley et al., 2025) propose a formal definition, called *projective composition*, of "correct" composition of multiple distributions (Du et al., 2023; Liu et al., 2022). In this paper, we specialize projective composition to a single conditional distribution to provide a precise definition of composition structure.

We develop a theoretical framework connecting compositional generalization with local mechanisms. Specifically, we generalize the concept of local scores to define *local conditional scores* (LCS): scores with *sparse dependencies* on both pixels and conditioners. That is, the score at a given pixel depends only on a subset of other pixels (such as a local neighborhood) and on only one or a few relevant conditioners (e.g. in the case of location-conditioning, only conditioners near the current pixel). We specialize projective composition (Bradley et al., 2025) to define a *conditional projective composition* (CPC) – a conditional distribution that is a projective composition of its own individual conditionals. We then prove an equivalence between conditional projective composition and local conditional scores at all noise levels. This equivalence means that locality in the score function is both necessary and sufficient for compositional factorization – observing one implies the other – and suggests that LCS may be a causal mechanism for compositional generalization, which we test via direct intervention. We extend this theoretical framework to relate

compositional structure and sparse score dependencies in feature-space (intuitively, concepts like style+content will compose in feature-space if the score of each 'style feature' depends only on a sparse set of style-related conditioners and other features).

We validate this theory through experiments. Returning to our location-conditioned CLEVR setting and comparing a model that we found to length-generalize with others that did not, we find that the length-generalizing model maintains pixel- and conditional-locality, while the non-length-generalizing models exhibit non-locality. We find that the correlation between length-generalization and conditional-locality also holds over a wider range of models with varying length-generalization. Further, we perform a direct causal intervention to test local conditional scores as a possible mechanism for composition generalization: we show that explicitly enforcing a local architecture enables length generalization in a model that previously failed. Finally, we present quantitative evidence of feature-space disentanglement in SDXL, connecting compositional structure with real-world text-to-image models.

## 2. Length generalization in CLEVR

In this section we study length generalization in conditional diffusion models trained on CLEVR datasets (Johnson et al., 2017), using a standard EDM2 U-net architecture (Karras et al., 2022) (details in Section E.1). Figure 1 shows length generalization or lack thereof in three location-conditioned models trained on CLEVR images with 1-3 objects. In **Experiment 1**, the location-conditioning labels *all* objects in the scene, using a 2d integer array representing a 2d grid over the image via the count of objects whose center falls

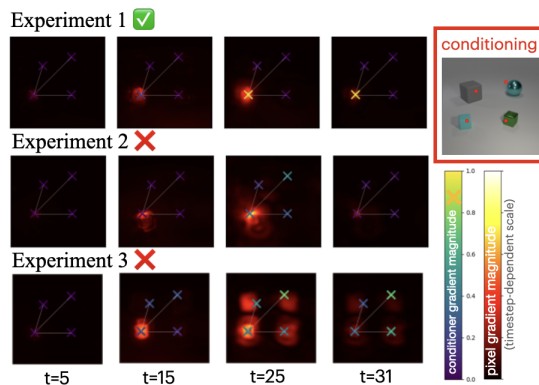

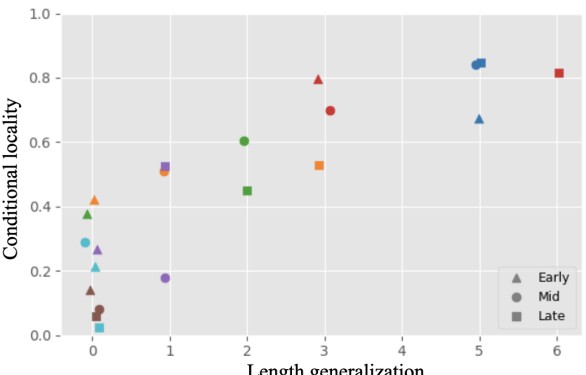

*Figure 2.* **Locality in location-conditioned CLEVR models.** For Experiments 1, 2 and 3 of Figure 1 each conditioned on four locations, we visualize pixel-locality via heatmaps, and conditional locality via the intensity of the × marker, centered at a pixel in the lower left, over a range of timesteps. (Section F describes the locality measurements; Figure 13 plots locality metrics; Figure 10 shows more pixel locations.) The length-generalizing Exp. 1 model exhibits strong pixel- and conditional-locality, while the non-length-generalizing Exp. 2 and 3 models both lack conditional-locality (the scores depend on non-local conditioners); Exp. 3 also lacks pixel-locality. These experiments support the theoretical equivalence between CPC and LCS.

*Figure 3.* **Length generalization vs. conditioner locality** for several models (different colors), each checkpointed early, mid, and late in training (different shapes). Details are in Section E.2.1. Length generalization and conditional locality are strongly correlated, and can emerge together over the course of training (e.g. orange, green, red models). Here, length-generalization ($x$-axis) is the number of locations to which the model can generalize ($K_{\max}$ in Table 1) *minus* the maximum number on which it was trained (e.g. +6 for a model trained on 1-3 locations that generalizes to 9). The conditional locality ($y$-axis) metric is described in Section F.

| Train data | Exp.1 | Exp.2 | Exp.3 | Exp.2L | Exp.3L |
|---|---|---|---|---|---|
| 1 object | 1 | 1 | 1 | 5 | – |
| 1-3 objects | 7 | 2 | 3 | 9 | 9 |
| 1-5 objects | 10 | 3 | 6 | 10 | – |

*Table 1.* **Upper limits of length generalization** in location-conditioned CLEVR. The table lists the maximum value, $K_{\max}$, such that the model "sometimes succeeds" for every $1 \leq K \leq K_{\max}$, as described in Section E.2. $K_{\max}$ is evaluated using an automated ResNet-18 classifier over 1024 samples per configuration. Results are shown for Exp. 1, 2, 3, 2L, 3L of Figure 1. Dashes indicate untested configurations.

within the grid cell (typically zero or one), as shown in Figure 7. We find that this model length-generalizes up to 7 objects. In **Experiment 2**, the setup is identical to that of Experiment 1 except that the conditioning only labels the location of a *single* randomly-selected object. Unsurprisingly, this model fails to meaningfully length-generalize beyond one conditioned location (note that when trained on 1-3 objects it may generate up to 2 "extra" objects at non-conditioned locations). **Experiment 3** conditions on the 2D locations of all objects using a list-style conditioner which places the (embedded) xy-locations of each object in an array padded with enough slots for up to 10 objects, with each location placed in a randomly chosen slot. This model fails to length-generalize beyond the 3 locations it was trained on. A priori, we should not "expect" length-generalization in any of these models. Although the data is naturally compositional, there is no guarantee that a model will learn a compositional structure from examples with

only 1-3 objects (which it could fit in many different ways). Evidently, the Experiment 1 model learns the underlying compositional structure of the data while the two other models do not. Table 1 gives a quantitative analysis of the limits of length generalization of the various location-conditioned models, trained on 1 object, 1-3 objects, or 1-5 objects, with samples shown in Figure 11. The table reports an approximate measure, $K_{\max}$, of the maximum number of objects to which each model can consistently length-generalize (details in Section E.2).

## 3. Theory: compositionality and locality

In this section we present theory connecting compositional generalization to generalized local mechanisms. We first define *local conditional scores* (LCS), an extension of local scores Kamb & Ganguli (2024); Niedoba et al. (2024) to account for flexible conditioners. An LCS evaluated at a given pixel has sparse dependencies on other pixels (generalizing local neighborhoods) and sparse dependencies on conditioners. Next, we define *conditional projective composition* (CPC), a special case of *projective composition* (PC) proposed in (Bradley et al., 2025) applied to a single conditional distribution. A conditional distribution that satisfies CPC factorizes into independent distributions over disjoint subsets of pixels that depend on a single condition. LCS and CPC are illustrated in Figure 4. We then prove that the score of a CPC is exactly an LCS: intuitively, *compositional distributions have local scores*. We verify this empirically in Figure 2, discussed further in Section 4. This result can be partially relaxed in an approximation that

**Conditional Projective Composition**

$$p(x|c_J) = p(x_{M_b^\complement}|\varnothing) \prod_{j \in J = \{0,1,2,3\}} p(x_{M_j}|c_j)$$

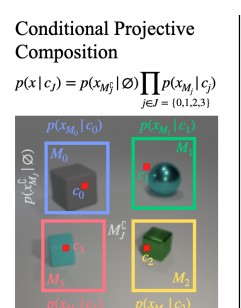

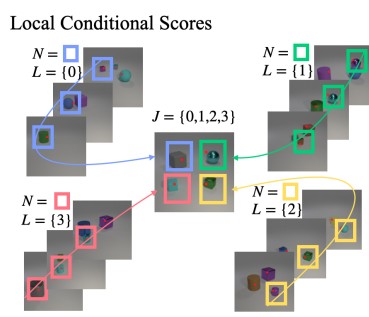

*Figure 4. Conditional projective composition* (CPC) and *local conditional scores* (LCS), illustrated for an image with four conditions $c_0, c_1, c_2, c_3$ (e.g. four object locations). A CPC factorizes the conditional distribution independently over disjoint pixel subsets $M_0, M_1, M_2, M_3, M_b$: the pixels in $M_j$ depend only on condition $c_j$, and the "background" pixels $M_b$ are unconditional. LCS is the corresponding property of the score function: the score at each pixel $i$ depends only on a subset $N_i$ of nearby pixels and a subset $L_i$ of relevant conditions. The CPC⇔LCS equivalence (Lemma 3.3) shows that these two perspectives – compositional structure in the distribution and locality in the score – are equivalent.

improves at higher noise, which could allow approximately-compositional structure to be resolved early in denoising. The theory can also be extended to *feature space*, to connect compositional structure (e.g. style+content) with sparse score dependencies (e.g. scores of 'style features' only depend on select conditioners and features relevant to style). The remainder of this section makes these claims precise.

**Background**   Our theory builds upon two lines of prior work: local scores and projective composition. Kamb & Ganguli (2024); Niedoba et al. (2024) propose that diffusion models learn local score functions, enabling them to generate samples not in the training set through mosaicing of local patches. Both works define local scores essentially as follows: let $x_N$ denote the restriction of $x$ to a subset of indices $N \subseteq [n]$, and $p(x_N)$ denote the marginal distribution of $p(x)$ on $x_N$. Intuitively, a local score at pixel $i$ depends only on a neighborhood of pixels centered at pixel $i$. That is, $s^t$ is a local score at time $t$ with neighborhood subsets $N_i^t$ (which may depend on the time $t$), if $s^t[x](i) := \nabla \log p^t[x_{N_i^t}](i)$, for all pixels $i$. However, these works study unconditional and class-conditional diffusion, not flexible conditioners central to compositional generalization. In this work, we will extend this concept to incorporate flexible conditioners and study compositionality.

(Bradley et al., 2025) introduce projective composition (PC) as a formal definition for "correctly composing" multiple distributions $\{p_b, p_1, p_2, \dots\}$, where $p_b$ is a "background" distribution and the $p_i$ are "concept distributions". One possible construction of a projective composition is given by $p_{\mathcal{J}}(x) := p_b(x_b) \prod_{j \in \mathcal{J}} p_j(x_{M_j})$, where the $M_j$ are dis-

joint subsets corresponding to the $p_j$, respectively. In this paper, we will specialize PC to define a specific compositional structure for conditional distributions. However, in contrast to Bradley et al. (2025)'s focus on explicit compositions, our goal is to connect compositional structure with score locality.

Next, we present new definitions and theory that build on local scores and projective composition.

**Local Conditional Scores**   We generalize the idea of local scores (Kamb & Ganguli, 2024; Niedoba et al., 2024) to account for flexible compositional conditioners,[1] which are central to compositional generalization. Let $p(x|c)$ be the true distribution over data $x \in \mathbb{R}^n$ conditioned on $c$. Let $x_N$ denote the restriction of $x$ to a subset of indices $N \subseteq [n]$, and $p(x_N|c)$ denote the marginal distribution of $p(\cdot|c)$ on $x_N$. Conditioners are represented as $c_{\mathcal{J}} = \{c_j, j \in \mathcal{J}\}$, where $\mathcal{J} \subseteq \mathcal{J}_{\text{all}}$ is a subset of all possible conditioners. We assume that $p$ is defined for any combination of conditioners, even those not seen during training. A local conditional score at pixel $i$, $s^t[x|c_{\mathcal{J}}](i)$, depends on two subsets (Figure 4): $N_i$, a subset of pixels relevant to pixel $i$, and $L_i(\mathcal{J})$, a subset of conditions in $\mathcal{J}$ relevant to pixel $i$. In general, the subsets $N_i$ and $L_i$ need not be disjoint (although this will later be necessary to achieve CPC equivalence), and may contain multiple objects or conditioners.

**Definition 3.1** (Local Conditional Score (LCS)).  We say that $s^t$ is a local conditional score at time $t$ with pixel subsets $N_i$ and conditional subsets $L_i^t$ (which may both depend on the time $t$), if

$$s^t[x|c_{\mathcal{J}}](i) := \nabla \log p^t[x_{N_i^t}|c_{L_i^t(\mathcal{J})}](i), \quad \forall i. \quad (1)$$

Importantly, Definition 3.1 does not strictly require "locality" but rather captures a "sparse dependency structure" where the score at index $i$ depends only on specific subsets $N_i$ and $L_i$. While $N_i$ is often a local neighborhood in image settings (and $L_i$ can be local e.g. for location-conditioners), these subsets can be arbitrary in general. We use "local" as an intuitive term for these sparse dependencies.

**Conditional Projective Composition**   To define a compositional structure for conditional distributions, we introduce conditional projective composition (CPC). We do not claim all distributions have this structure, but we will show that those that do also have a local score structure (LCS), suggesting a mechanism for compositional generalization. We specialize (Bradley et al., 2025)'s pixel-space projective composition to the case where the concept distributions

---

[1]Our definition breaks slightly from the originals in defining local scores via a distribution rather than finite training set, and also omits equivariance, which is unnecessary for our theory.

$p_j$ represent a *single* conditional distribution $p(x|c_j)$ conditioned on different $c_j$, and the background distribution $p_b(x)$ is $p(x|\emptyset)$ (i.e., with no conditioners active).

**Definition 3.2** ((Pixel-space) Conditional Projective Composition (CPC)). We say that $p(x|c)$ is a conditional projective composition if there exist disjoint sets $M_j$ for all conditions $j \in \mathcal{J}_{\text{all}}$ such that, for any set of conditions $\mathcal{J} \in \mathcal{J}_{\text{all}}$, $p(x|c_{\mathcal{J}})$ decomposes as

$$p(x|c_{\mathcal{J}}) := p(x_{M_{\mathcal{J}}^{\complement}}|\emptyset) \prod_{j \in \mathcal{J}} p(x_{M_j}|c_j), \qquad (2)$$

where $M_{\mathcal{J}}^{\complement} := \mathbb{R}^n \setminus \cup_{j \in \mathcal{J}} M_j$ denotes the set of pixels not controlled by any active condition.

This definition means that the conditional distribution $p$ decomposes into independent marginal distributions $p(x_{M_j}|c_j)$, each depending only on subset $M_j$ and condition $c_j$, as shown in Figure 4. That is, $p$ modifies $M_j$ according to $c_j$ independently of other pixel sets and conditioners. This condition is quite strong, but we will partially relax it in our theory.

## 3.1. Equivalence between compositional structure and local scores

In this section we present theory showing that the score of an (approximately) conditional projective composition is (approximately) a particular local conditional score. We begin by showing that a specific local conditional score is *exact* for a conditional projective composition.

**Lemma 3.3** (Equivalence of CPC and LCS). *A distribution $p(x|c_{\mathcal{J}})$ is a pixel-space Conditional Projective Composition (Definition 3.2) with disjoint sets $\{M_j\}_{j \in \mathcal{J}_{all}}$ if and only if its score $s(x|c_{\mathcal{J}})$ is a Local Conditional Score (Definition 3.1) with subsets*

$$L_i^t(\mathcal{J}) = \begin{cases} \{j\} \cap \mathcal{J}, & \text{if } i \in M_j \\ \emptyset, & \text{else,} \end{cases} \quad \text{and}$$

$$N_i^t = \begin{cases} M_j, & \text{if } i \in M_j \\ M_b, & \text{else,} \end{cases} \quad \text{where } M_b := M_{\mathcal{J}_{all}}^{\complement}.$$

The proof is in Section B. The lemma says that when the locality structure of $s^t$ is precisely connected to the compositional structure of $p^t$ – that is, if pixel $i$ belongs to the subset $M_j$ controlled by condition $j$ in the CPC, then $L_i = \{j\}$ (pixel $i$ only depends on condition $j$), and $N_i = M_j$ (pixel $i$ only depends on pixels in $M_j$) – then $s^t$ is exactly the score of $p^t$. Thus, there is an *equivalence* between CPCs and LCSs. This is illustrated in Figure 4.

This equivalence has two important implications. First, it means that locality in the score function is both necessary and sufficient for compositional factorization: if a trained model exhibits local conditional scores, it has implicitly learned a compositional structure, and vice versa. Second, beyond this equivalence, we are interested in whether LCS could be a *causal mechanism* for compositional generalization. The CPC⇔LCS equivalence is necessary but not sufficient to establish causality; the causal intervention experiments in Section 4 provide direct evidence that enforcing LCS can *cause* compositional generalization. We note that a related CPC⇔LCS-type connection has been implicitly leveraged in the object-centric representation learning literature, where block-diagonal Jacobians (analogous to LCS) and slot-based generative models (analogous to CPC) have been shown to support compositional generalization in autoencoders (Brady et al., 2023; Wiedemer et al.; Brady et al., 2024); our work makes this equivalence explicit and proves it in the conditional score function setting (see Section A).

**A relaxation** What about imperfect compositionality? We can relax Lemma 3.3 to show that the score of an *approximately* CPC distribution is *approximately* an LCS. Further, we show that the CPC approximation becomes more accurate – intuitively, distributions are "more compositional" – at higher noise. The precise statements and proofs are given in Section C. Why might this be helpful? If conditional dependencies are strongest at high noise and pixel dependencies take over at low noise (as we observe in Figure 2), then local conditional mechanisms might be able to establish large-scale compositional structure (like object count and location) early in denoising, leaving less-compositional details to be resolved at low noise via local unconditional denoising.

## 3.2. Feature-space conditional projective composition

What if CPC does *not* hold in pixel-space – as is typically the case for non-location conditioners, like the color-conditioning of Figure 14, or text-to-image prompts such as "a watercolor of a cat eating sushi with chopsticks"? Pixel-space CPC is unlikely since each condition potentially affects many pixels (e.g., "watercolor" style would apply to all pixels). In these cases, we hypothesize that the local unconditional denoising mechanism still applies at low noise (Kamb & Ganguli, 2024; Niedoba et al., 2024). But is there any hope of compositional generalization at high noise?

It follows directly from Lemma 3.3 that if a distribution has a CPC structure *in feature-space* then its score is an LCS *in feature-space* (we name these F-CPC/F-LCS, respectively):

**Corollary 3.4** (F-LCS is exact for F-CPC; informal). *Suppose that $p(x|c)$ is an F-CPC (a CPC in feature-space): that is, $\mathcal{A}\sharp p(z|c)$ is a CPC, where $\mathcal{A}$ is an invertible transform, and $z := \mathcal{A}(x)$ is the feature-space representation. Then the feature-space score $\nabla_z \log(\mathcal{A}\sharp p)^t(z|c)$ is an F-LCS (an LCS in feature-space) with neighborhoods $N_i, L_i$ related to the CPC subsets $\{M_j\}$ of $\mathcal{A}\sharp p$ as in Lemma 3.3.*

For example, in an appropriate feature-space, the concepts "watercolor", "cat", and "sushi" might have F-CPC structure – despite interacting in pixel space. (Note that an F-LCS will usually have sparse-dependencies rather than literal "locality", as allowed by Definition 3.1, since interacting features need not be contiguous). However, challenges remain: if the scores are F-LCS in some feature-space, in order to exploit the sparse structure the model must *learn* this feature-space mapping and its inverse, in addition to the local subsets, making the learning process significantly more challenging.[2] This argument is made precisely in Section D.

As a practical heuristic for identifying F-LCS structure, we propose an empirically-testable necessary-but-not-sufficient condition for F-LCS, based on orthogonality between score differences (similar to Bradley et al. (2025) Lemma 8.1). The proof is in Section D.1.

**Lemma 3.5** (F-LCS necessary-but-not-sufficient heuristic). *Let $s_{\mathcal{A}}^t(z|c) := \nabla_z \log(\mathcal{A} \sharp p)^t(z|c)$ be an F-LCS score in a feature-space given by transform $\mathcal{A}$, and let $t_{\max}$ denote the highest noise level. Then:*

$$d_i^T d_j = 0 \quad \forall i \neq j, \quad d_i := \mathbb{E}[s_{\mathcal{A}}^{t_{\max}}(\cdot|c_i)] - \mathbb{E}[s_{\mathcal{A}}^{t_{\max}}(\cdot|\emptyset)]$$

*where the expectation is over the feature-space transformed noise distribution $\mathcal{A} \sharp \mathcal{N}(0, \sigma_{t_{\max}})$.*

Practically, to study a feature-space represented within layer $\ell$ of a denoising network, we compute $s_{\mathcal{A}}^{t_{\max}}(\cdot|c)$ by drawing a noise sample, running the first denoising step (at time $t_{\max}$) to compute the conditional score, and hooking the activation of layer $\ell$. To obtain $d$ we average over multiple noise samples and compute the conditional-unconditional difference. Finally, we can construct a cosine similarity matrix $\{d_i d_j / \|d_i\| \|d_j\|\}_{i,j}$: low similarity off-diagonal is evidence of F-LCS.

*Remark* 3.6. F-CPC/F-LCS structure should be viewed as a type of *disentanglement*, on which there is a rich literature (Bengio et al., 2013; Higgins et al., 2017; Chen et al., 2018; Kim & Mnih, 2018; Locatello et al., 2019; Kotovenko et al., 2019; Locatello et al., 2019; Watters et al., 2019; Yang et al., 2023; Zhang et al., 2023). To quote Karras et al. (2019): "There are various definitions for disentanglement, but a common goal is a latent space that consists of linear subspaces, each of which controls one factor of variation." F-CPC/F-LCS satisfies this definition with the additional requirement that the subspaces be orthogonal. Intuitively, disentanglement is often thought to promote compositionality; our specific definitions and theory of F-CPC/F-LCS makes this connection precise and provable.

*Remark* 3.7. Identifying feature-space disentanglement is fundamentally difficult since independence between high-dimensional random variables cannot be tested in polynomial time. However, a variety of practical metrics have been proposed (Higgins et al., 2017; Kim & Mnih, 2018; Chen et al., 2018); (Locatello et al., 2019) shows that many common metrics are fairly correlated with each other. The heuristic of Lemma 3.5 is part of the broader family of closely-related disentanglement metrics, but is specifically designed to test F-CPC/F-LCS.

## 4. Additional Experiments

Our theory shows an equivalence between LCS and CPC (which implies length generalization). We test this directly in location-conditioned CLEVR models, where the compositional structure holds in pixel-space, and location-conditioners possess a direct notion of locality. Further, we test whether LCS could be a *causal* mechanism via a direct intervention: enforcing an explicitly LCS architecture to "fix" length generalization that previously failed. We also show partial length generalization in color-conditioned CLEVR. Finally, we investigate local/compositional structure in both pixel- and feature-space in SDXL, providing quantitative evidence for local conditional scores in feature-space.

**Pixel-space locality in location-conditioned CLEVR** Figure 2 shows pixel- and conditional-locality in Experiments 1, 2, and 3. We first observe that Exp. 1 and 2 maintain pixel-locality at both low and high noise levels, contrasting with prior empirical findings on datasets like CIFAR-10 (Kamb & Ganguli, 2024; Niedoba et al., 2024) – reproduced in Figure 12 – showing delocalization at high noise. The difference likely stems from CLEVR images containing multiple, nearly-independent objects; unlike the datasets with a single centered subject studied in prior work. The non-length-generalizing Exp. 3 model lacks pixel-locality. Second, we note significant differences in conditional-locality between Exp. 1 vs. 2 and 3. The length-generalizing Exp. 1 model exhibits strong conditional-locality at high noise, while the non-length-generalizing Exp. 2 and 3 models lack conditional-locality at high noise (scores near conditioned locations either fail to respond to any conditions or depend on several non-local conditions). At low noise, all models transition to pixel-local *unconditional* denoising, as in Kamb & Ganguli (2024); Niedoba et al. (2024). These experiments support our prediction that length generalization depends on pixel- and conditional-locality, and suggest that conditional-locality at high noise plays a particularly important role. Locality metrics are plotted in Figure 13, Figure 10 shows additional pixel locations, and Section F details the locality measurements. Figure 3 plots length generalization vs. conditioner locality for several models (different colors),

---

[2]In fact, even learning the feature-space transform and its inverse for the noiseless distribution is not enough: the model technically needs to learn a *noise-dependent* feature-map, since pixel-space noise does not commute with the feature-space transform.

each checkpointed early, mid, and late in training (different shapes). Details are in Section E.2.1. Length generalization and conditional locality are strongly correlated and can emerge together over the course of training.

**A causal intervention in location-conditioned CLEVR** Experiment 2L (Figure 1) tests our theory via a direct causal intervention, wherein we design a local model architecture that explicitly enforces the local conditional scores defined in our theory. Specifically, we modify the architecture to process images as a grid of independent patches, each conditioned on the subset of locations within the patch. This design is conceptually related to local model architectures proposed in prior works like (Li et al., 2023; Zheng et al., 2023; Cheng et al., 2023; Watters et al., 2019). We train the local model using the same conditioning as the failing Exp. 2 (labeling only a single object location), and find that the local architectural intervention causes it to length-generalize (Table 1). In fact, the local model trained on *only one object* can length-generalize up to 5 locations (Figure 9, Table 1). This supports the hypothesis that local conditional scores could be a *causal* mechanism for compositional generalization. Locality metrics and plots for Exp. 2L are shown in Figure 13 and 10. Additional details in Section E.3. We perform an analogous local intervention for Experiment 3 that similarly improves length-generalization as described in Section E.4.

**Length generalization in color-conditioned CLEVR** In Section E.5 (Figure 14 and Table 4), we explore length generalization in color-conditioned CLEVR, where we may expect compositional structure to exist in an appropriate feature space. We observe that length generalization is possible to some extent – e.g. a model trained on 1-5 objects can generalize to 7 colors.

**SDXL investigation** Can local mechanisms help to explain compositional generalization in real-world diffusion text-to-image models? While length generalization served as a controlled and verifiable special-case of compositional generalization in our CLEVR experiments, we now return to the broader question of compositions of novel combinations of concepts. We begin to explore this question by studying compositional/local structure in both pixel- and feature-space in a pretrained ('out-of-the-box') SDXL model (Podell et al., 2023). In Figure 5, we investigate pixel-space locality using a prompt containing implicit location information ("middle", "right", "left"). We measure pixel-locality via pixel gradient magnitude (Jacobian heatmaps) and conditional-locality via per-word ablation influence, where each word's token embedding is replaced with padding and the resulting change in the score is measured (averaged over 10 seeds; see Section G&F for details). We find that pixel-locality is present, particularly at low

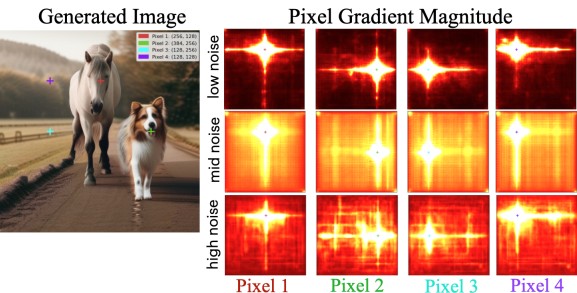

*Figure 5.* **SDXL pixel-space locality.** (Left) An SDXL-generated image for the prompt "a beautiful photograph with a horse in the middle, a dog on the left, and a cat on the right," with four analysis locations marked. (Right) Pixel gradient magnitude heatmaps at low, mid, and high noise for each location, showing pixel-locality (localized cross patterns) especially at low noise. Per-word conditional influence analysis (Figure 15) reveals that conditional-locality is largely absent in pixel-space. Averaged over 10 seeds; additional seeds and prompts in Figure 16, spatial word-influence heatmaps in Figure 17.

noise: the Jacobian heatmaps show localized cross-shaped patterns centered on each analysis pixel (reflecting the effective receptive field of the convolutional U-Net architecture (Luo et al., 2016)). However, conditional-locality is largely absent in pixel-space: the per-word influence profiles are nearly identical at different pixel locations (Figure 15), and spatial word-influence heatmaps are largely spatially uniform (Figures 16 and 17). Consistent with this lack of pixel-space conditional-locality, SDXL rarely succeeds in correctly composing all three animals at the specified locations (Figure 16). This lack of pixel-space conditional-locality motivates looking beyond pixel-space.

Next, we move beyond pixel-space to study F-LCS structure within the learned feature-spaces of SDXL. Certain concepts, such as animals vs. art styles, almost certainly interact in pixel-space but might be disentangled in the network's internal representation. In fact, there is significant evidence suggesting concept disentanglement (according to various metrics) within the learned feature-spaces of large-scale diffusion models (Karras et al., 2019; Kotovenko et al., 2019; Gatys et al., 2016; Zhu et al., 2017), which may help to explain their compositional abilities. To connect directly with our theory, we measure our F-LCS disentanglement heuristic given by Lemma 3.5, and connect this with compositional generalization. Specifically, we extract hidden-state activations (post-FFN outputs of each `BasicTransformerBlock`) from the SDXL U-net for 24 prompts spanning 4 semantic categories (8 animals, 7 art styles, 6 foods, 3 accessories), with 10 seeds per prompt. We compute the Lemma 3.5 heuristic (cosine similarity of mean activation differences relative to an unconditional baseline) and find quantitative evidence of F-LCS structure: within-category cosine similarities (0.41–0.47) substantially exceed between-category similarities (0.27–0.33),

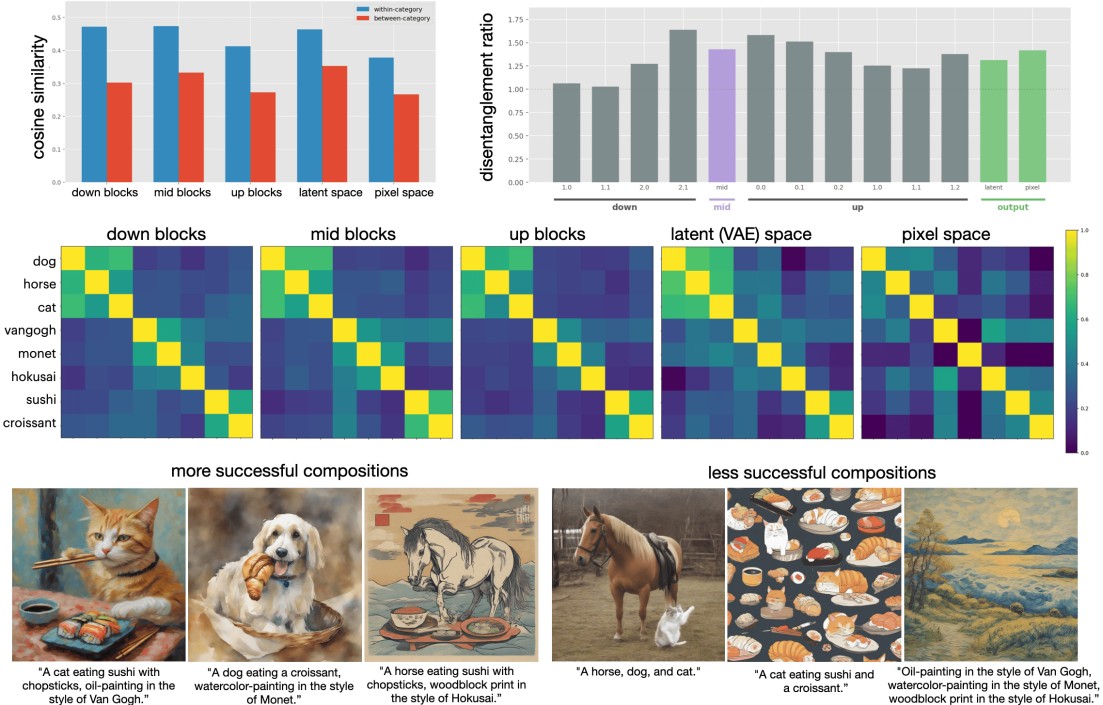

*Figure 6.* **Feature-space disentanglement in SDXL.** (Top left) Within- vs. between-category mean cosine similarity (F-LCS heuristic, Lemma 3.5) for hidden-state activations in the down, mid, and up blocks (at high noise) and for output spaces (latent/VAE, pixel). Feature-space representations are more disentangled (larger within/between gap) than output spaces. (Top right) Per-layer disentanglement ratio (within/between cosine similarity) across all transformer layers, showing an arch-shaped profile peaking near the U-net bottleneck. (Middle) $8 \times 8$ cosine similarity heatmaps for 8 representative concepts across feature-space (down, mid, up blocks) and output spaces (latent, pixel). Block-diagonal category structure (animals, art styles, foods) is most pronounced in feature-space. (Bottom) Example SDXL compositions: concepts that are disentangled in feature-space compose more successfully. Methodology: 24 prompts across 4 categories, 10 seeds, hidden states (post-FFN activations of transformer blocks) at high noise; see Section G and Figure 18.

yielding disentanglement ratios of 1.4–1.6×. Feature-space is more disentangled than both pixel-space (ratio 1.42) and latent/VAE-space (ratio 1.31). The per-layer disentanglement profile peaks near the U-net bottleneck (last encoder and first decoder layers), and disentanglement is strongest at high noise levels. Details in Section G.

One limitation of this study is that since SDXL's training data is undisclosed, we cannot be certain about which compositions are truly out of distribution. To address this, we also validated our findings on a Matryoshka Diffusion Model (Gu et al., 2023a) trained on the known Flickr dataset (Section H). We ran the same hidden-state disentanglement analysis on MDM's innermost U-net with 12 single-concept prompts and found qualitatively similar F-LCS structure: disentanglement ratios of 1.49–1.68× across all blocks (Figure 19). This controlled setting confirms that compositional generalization (e.g., 'a cow jumping over a candlestick') emerges even when the specific combination is definitely absent from the training set.

## 5. Discussion and future work

Whether local mechanisms can explain compositional generalization in real-world diffusion models remains an open question. The preliminary SDXL experiments in Section 4 are meant only to be suggestive, and much more work is needed to fully understand compositional generalization in modern text-to-image models. This setting presents several challenges. First, since we often don't know what was in the training set, it is unclear which prompts are actually OOD (however, see Section H for a small exploratory study of a model trained on a known dataset). Second, some kinds of compositional structure, such as style+content, exist only in feature-space, requiring more complex studies of the model's learned representation as in Figures 14 and 6, potentially relying on heuristics such as Lemma 3.5. Despite significant evidence for the existence of disentangled feature spaces (Chen et al., 2018; Kim & Mnih, 2018; Yang et al., 2023; Locatello et al., 2019; Zhang et al., 2023; Ilharco et al., 2022) and diffusion models' ability to learn them in some cases (Karras et al., 2019; Kotovenko et al., 2019; Gatys et al., 2016; Zhu et al., 2017), disentanglement in diffusion representations is still not fully understood. Future work

could also exploit our finding of local conditional scores as a mechanism to improve compositional generalization. Our causal intervention in Experiment 2L shows that in a simple setting, enforcing an explicitly local architecture can improve generalization, suggesting that similar architectural, training, or inference-based interventions might be able to improve real-world models. Several existing methods can be viewed as implicit "local interventions": for instance, layout-to-image methods that use explicit local constraints or biases (see Section A), or more generally, sparse attention architectures (Child et al., 2019; Sun et al., 2022). Our theory helps to explain why these approaches are beneficial, and suggests more precise ways to target compositional generalization by specifically enforcing local conditional scores. When compositional structure exists only in feature-space, such interventions become more complex. The challenge becomes two-fold: we must first identify or attempt to induce feature-space transformations revealing the compositional structure, and then apply local interventions within the learned feature space: perhaps via sparsity-inducing regularization (such as L1) or explicitly-sparse architectures. Larger-scale studies on complex, real-world datasets are essential to clarify the challenges and explore opportunities to improve compositional generalization via local interventions.

## 6. Conclusion

We proposed local conditional scores as a possible mechanism for compositional generalization. Theoretically, we proved an equivalence between conditional projective composition (a specific compositional structure) and local conditional scores (which capture both pixel- and conditional-locality); this theory extends to feature-space compositionality. Empirically, we verified that length generalization in location-conditioned CLEVR models corresponds with local conditional dependencies at high noise combined with pixel-locality at low noise. Then, we demonstrated through a causal intervention that enforcing a local architecture restores length generalization in a model that previously failed. We also observed length generalization in color-conditioned CLEVR, and preliminary evidence of compositional structure in both pixel- and feature-space in SDXL. Our results support local conditional scores as a potential mechanism of compositional generalization in conditional diffusion models, offering a lens to understand when and how generalization is achieved, and potential avenues to improve it.

## Acknowledgements

Many thanks to Josh Susskind, Preetum Nakkiran, and Miguel Angel Bautista for helpful discussions and feedback throughout this project, and David Berthelot and James Thornton for developing code infrastructure supporting the CLEVR experiments.

## Impact Statement

This paper presents work whose goal is to advance the field of Machine Learning. There are many potential societal consequences of our work, none which we feel must be specifically highlighted here.

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

# A. Additional Related Work

**Locality and generalization in diffusion models**    Kamb & Ganguli (2024); Niedoba et al. (2024) argue that "creativity" in unconditional and class-conditional models arises from a bias toward learning local denoisers, while Lukoianov et al. (2025) challenge some of the conclusions of Kamb & Ganguli (2024) by arguing that locality arises from statistical properties of the data rather than network inductive bias. Our work builds on the idea of locality by showing that it can also be a mechanism for compositional generalization in conditional models, which none of the previous works addressed. We are agnostic about whether locality arises from data statistics or inductive bias in practice (likely both are at play); our main insight is that compositional generalization can be achieved when the network learns a local – or equivalently, compositional – structure by any means. (Our Experiment 2L suggests that local architectural interventions can promote learning the underlying compositional structure of the data, but this structure can also be learned "naturally" as in Experiment 1.) Note also that Kamb & Ganguli (2024) propose equivariance while Niedoba et al. (2024); Lukoianov et al. (2025) omit or argue against it; in our theory and experiments equivariance is not necessary.

**Generalization of diffusion models**    Kadkhodaie et al. (2023) propose shrinkage in a geometry-adaptive harmonic bias as a mechanism for generalization; in the framework of our theory, this can be thought of as a bias toward sparse dependencies in a particular feature-space. Gu et al. (2023b) make an experimentally study of potential causes generalization in unconditional and class-conditional settings based on the characteristics of the dataset and choices for training and model. Bertrand et al. (2025) give empirical evidence that for flow models in high-dimensions, generalization arises primarily from network inductive biases rather than noise in the flow-matching loss.

**Learning Dynamics of Composition**    An interesting line of work focuses on the learning dynamics of models trained on compositional data. Okawa et al. (2024); Park et al. (2024) demonstrate sudden emergence of compositional generalization in controlled synthetic settings where they test novel combinations of attributes of a single object (e.g. blue square, red triangle $\rightarrow$ blue triangle?). Sclocchi et al. (2025); Favero et al. (2025) study learning dynamics in hierarchical models, showing that higher-level features take longer to learn.

**Explicit composition of multiple diffusion models**    Du & Kaelbling (2024); Liu et al. (2022); Bradley et al. (2025) study *explicit* compositions of multiple diffusion models via linear score combination, demonstrating length-generalization in CLEVR in some cases. Bradley et al. (2025) proposed Projective Composition as a definition of "correct" compositions of multiple models; here we use it to precisely characterize compositionality in a single conditional model.

**Layout-to-Image Diffusion Models**    Our theory may help to explain the success of layout-to-image methods that use architectural locality priors (Li et al., 2023; Zheng et al., 2023; Cheng et al., 2023; 2024) or inference-time locality constraints (Dahary et al., 2024; Xue et al., 2023; Xie et al., 2023) to improve multi-object generation. Although these works primarily report improved grounding and controllability rather than explicit OOD composition, their interventions can be viewed as approximate causal tests, where increased locality leads to improved multi-object behavior, consistent with our theory and similar to the causal intervention Exp. 2L.

**Object-centric representations and compositional generalization**    A related line of work studies compositional generalization in object-centric autoencoders. Brady et al. (2023) establish conditions under which autoencoders can provably recover object-centric (slot-based) representations, where the generative model has a factored structure analogous to our CPC. Wiedemer et al. prove that object-centric autoencoders with block-diagonal Jacobians – i.e., each output pixel depends only on the latent slot of its corresponding object, a condition analogous to our LCS – can compositionally generalize to novel object combinations. Their disentangled latent representations correspond to our feature-space CPC/LCS (F-CPC/F-LCS). Brady et al. (2024) extend these ideas to a general principle of "interaction asymmetry": within-object interactions are stronger than between-object interactions, analogous to the locality structure in our LCS where nearby pixels (within an object) have strong score dependencies while distant pixels (across objects) have weak or zero dependencies. While these works focus on deterministic autoencoders rather than conditional diffusion models, they identify a closely related structural connection between compositional factorization and sparse Jacobians, implicitly leveraging a CPC$\Leftrightarrow$LCS-type equivalence. Our work makes this equivalence explicit and proves it in the probabilistic score function setting.

**Feature-space disentanglement**    There is a large body of work towards designing disentanglement metrics appropriate for "real-world" distributions (e.g. disentanglement metrics introduced by BetaVAE (Higgins et al., 2017), FactorVAE (Kim &

Mnih, 2018); MIG in (Chen et al., 2018), etc.). Many works have shown evidence of disentanglement over a variety of datasets such as CelebA (Karras et al., 2019; Chen et al., 2018; Kim & Mnih, 2018), Shapes3D (Locatello et al., 2019), and dSprites (Watters et al., 2019; Chen et al., 2018); also (Kotovenko et al., 2019) explores disentanglement between style and content for style transfer. Other works have shown that diffusion models have some ability to learn disentangled feature spaces (Karras et al., 2019; Kotovenko et al., 2019; Gatys et al., 2016; Zhu et al., 2017). Nevertheless disentanglement in diffusion representations is still not fully understood.

# B. Proof of Lemma 3.3

*Proof.* (Lemma 3.3)

First, we prove that CPC $\implies$ LCS.

Let $p$ be the CPC given by

$$p(x|c_{\mathcal{J}}) := p(x_{M_{\mathcal{J}}^{\complement}}|\emptyset) \prod_{j \in \mathcal{J}} p(x_{M_j}|c_j), \quad \forall \mathcal{J} \in \mathcal{J}_{\text{all}}$$

Since convolution with isotropic Gaussian noise acts independently on orthogonal subspaces, this factorization is preserved at any diffusion time $t$ (Bradley et al., 2025). Thus, the time-dependent distribution is:

$$p^t(x|c_{\mathcal{J}}) := p^t(x_{M_{\mathcal{J}}^{\complement}}|\emptyset) \prod_{j \in \mathcal{J}} p^t(x_{M_j}|c_j), \quad \forall \mathcal{J}$$

Let $s$ be the LCS given by

$$s^t[x|c_{\mathcal{J}}](i) := \nabla \log p^t[x_{N_i^t}|c_{L_i^t(\mathcal{J})}](i), \quad \forall i, \forall t$$

$$L_i^t(\mathcal{J}) = \begin{cases} \{j\} \cap \mathcal{J}, & \text{if } i \in M_j \\ \emptyset, & \text{else} \end{cases} \qquad N_i^t = \begin{cases} M_j, & \text{if } i \in M_j \\ M_b, & \text{else,} \end{cases} \quad \text{where } M_b := M_{\mathcal{J}_{\text{all}}}^{\complement}.$$

We want to show that $s$ is exactly the score of $p$:

$$s^t(x|c_{\mathcal{J}}) = \nabla \log p^t(x|c_{\mathcal{J}}), \quad \forall \mathcal{J}.$$

To see this, we first analyze $p^t$ at each pixel $i$:

$$\nabla \log p^t(x|c_{\mathcal{J}}) := \nabla \log p^t(x_{M_{\mathcal{J}}^{\complement}}|\emptyset) + \sum_{j \in \mathcal{J}} \nabla \log p^t(x_{M_j}|c_j), \quad \forall \mathcal{J}$$

$$\nabla \log p^t(x_{M_j}|c_j)(i) = 0, \quad \forall i \notin M_j, \quad \text{since } p^t(x_{M_j}) \text{ does not depend on } x_i$$

$$\implies j \in \mathcal{J}, \quad i \in M_j \implies \nabla \log p^t(x|c_{\mathcal{J}})(i) = \nabla \log p^t(x_{M_j}|c_j)(i)$$

$$j \notin \mathcal{J}, \quad i \in M_j \implies \nabla \log p^t(x|c_{\mathcal{J}})(i) = \nabla \log p^t(x_{M_j}|\emptyset)(i)$$

$$i \in M_b \implies \nabla \log p^t(x|c_{\mathcal{J}})(i) = \nabla \log p_b^t(x_{M_b}|\emptyset)(i).$$

Next we analyze $s$ at each pixel $i$:

$$s^t[x|c_{\mathcal{J}}](i) := \nabla \log p^t[x_{N_i^t}|c_{L_i^t(\mathcal{J})}](i)$$

$$j \in \mathcal{J}, \quad i \in M_j \implies L_i^t(\mathcal{J}) = \{j\}, \quad N_i^t = M_j \implies s^t[x|c_{\mathcal{J}}](i) = \nabla \log p^t(x_{M_j}|c_j)(i)$$

$$j \notin \mathcal{J}, \quad i \in M_j \implies L_i^t(\mathcal{J}) = \emptyset, \quad N_i^t = M_j \implies s^t[x|c_{\mathcal{J}}](i) = \nabla \log p^t(x_{M_j}|\emptyset)(i)$$

$$i \in M_b \implies L_i^t = \emptyset, \quad M_b \subseteq N_i^t \implies s^t(x|c_{\mathcal{J}})(i) = \nabla \log p_b^t[x_{M_b}|\emptyset](i).$$

Comparing $s^t(x|c_{\mathcal{J}})(i)$ and $\nabla \log p^t(x|c_{\mathcal{J}})(i)$ in each of the three cases, we see that $s^t[x|c](i) = \nabla \log p^t(x|c)(i)$ for all pixels $i$, hence

$$s^t(x|c_{\mathcal{J}}) = \nabla \log p^t(x|c_{\mathcal{J}}), \quad \forall \mathcal{J}.$$

Next, we prove the converse LCS $\implies$ CPC.

Assume $s$ is an LCS with neighborhoods $N_i = M_j$ and $L_i = \{j\}$ for $i \in M_j$. By definition, for any pixel $i \in M_j$, the score component $s_i(x|c_{\mathcal{J}})$ depends only on $x_{M_j}$ and $c_j$.

Consider the Hessian of the log-density, $H_{ik} = \frac{\partial^2 \log p}{\partial x_i \partial x_k} = \frac{\partial s_i}{\partial x_k}$. If $i \in M_j$ and $k \in M_l$ with $j \neq l$, then $k \notin N_i$. By the LCS assumption, $s_i$ is independent of $x_k$, implying:

$$\frac{\partial^2 \log p}{\partial x_i \partial x_k} = 0 \quad \forall i \in M_j, k \in M_l, j \neq l. \tag{3}$$

Since the mixed partial derivatives between disjoint sets vanish, the log-density is additively separable:

$$\log p(x|c_{\mathcal{J}}) = \sum_{j \in \mathcal{J}} \phi_j(x_{M_j}, c_{\mathcal{J}}) + C, \quad \text{with } s_i = \nabla_{x_i} \phi_j(x_{M_j}, c_{\mathcal{J}}) \text{ if } i \in M_j \tag{4}$$

By the LCS assumption, $s_i$ depends only on $c_j$ (not the full set $c_{\mathcal{J}}$), so $\phi_j$ depends on conditioning only via $c_j$ (up to a constant). Thus:

$$\log p(x|c_{\mathcal{J}}) = \sum_{j \in \mathcal{J}} \tilde{\phi}_j(x_{M_j}, c_j) + \text{const.} \tag{5}$$

Exponentiating yields $p(x|c_{\mathcal{J}}) \propto \prod_{j \in \mathcal{J}} \exp(\tilde{\phi}_j(x_{M_j}, c_j))$, which implies $p$ factorizes into independent marginals $p(x_{M_j}|c_j)$. Thus, $p$ is a CPC.

$\square$

## C. Relaxation of Lemma 3.3

In this section we provide a collection of lemmas showing that the score of an approximately CPC distribution is approximately an LCS, and that this approximation becomes more accurate as noise increases. We state all lemmas first and then provide the proofs at the end.

We begin by defining a notion of *approximate* conditional projective composition.

**Definition C.1** (Approximate CPC). We say that a conditional distribution $p^t(x|c)$ is approximately-CPC with errors $\{\varepsilon_j, \tilde{\varepsilon}_j, \varepsilon_b\}$ if

$$\sup_{x_{M_j^{\mathsf{c}}}} \text{KL}[p(x_{M_j}|c_{\mathcal{J}}, x_{M_j^{\mathsf{c}}}) || p(x_{M_j}|c_j)] \leq \varepsilon_j, \quad \forall \mathcal{J}, \quad \forall j \in \mathcal{J} \tag{6}$$

$$\sup_{x_{M_j^{\mathsf{c}}}} \text{KL}[p(x_{M_j}|c_{\mathcal{J}}, x_{M_j^{\mathsf{c}}}) || p(x_{M_j}|\emptyset)] \leq \tilde{\varepsilon}_j, \quad \forall \mathcal{J}, \quad \forall j \notin \mathcal{J} \tag{7}$$

$$\sup_{x_{M_b^{\mathsf{c}}}} \text{KL}[p(x_{M_b}|c_{\mathcal{J}}, x_{M_b^{\mathsf{c}}}) || p(x_{M_b})] \leq \varepsilon_b, \quad \forall \mathcal{J}. \tag{8}$$

The following lemma is relaxation of Lemma 3.3. It shows that the score of an *approximately-CPC* distribution is *approximately* an LCS.

**Lemma C.2** (LCS approximates score of approximate-CPC). *Let $p^t(x|c)$ be approximately-CPC with errors $\{\varepsilon_j, \tilde{\varepsilon}_j, \varepsilon_b\}$ per Definition C.1. Define a local conditional score $s$ as in Lemma 3.3, and let $\widehat{p}$ be the induced distribution s.t. $s^t(x|c_{\mathcal{J}}) = \nabla \log \widehat{p}^t(x|c_{\mathcal{J}})$. Then*

$$KL(p(\cdot|c_{\mathcal{J}}) || \widehat{p}(\cdot|c_{\mathcal{J}})) \leq \sum_{j \in \mathcal{J}} \varepsilon_j + \sum_{j \notin \mathcal{J}} \tilde{\varepsilon}_j + \varepsilon_b.$$

Further, we can show that the approximation errors in Definition C.1 decrease as noise is added: intuitively, an approximately-compositional distribution gets *more compositional* as noise is added.

**Lemma C.3.** *Suppose that the supremum of* $\sup_y [KL(N_t[p](x)||N_t[p](x|y))]$ *is attained for all $t$. Then*

$$\sup_y [KL(N_t[p](x)||N_t[p](x|y))]$$

*is decreasing in $t$.*

The proof of Lemma C.3 essentially follows from the fact that adding Gaussian noise decreases the KL divergence between distributions:

**Lemma C.4** (Standard; KL divergence decreases with noise)**.**

$$\frac{\partial}{\partial t} KL(N_t[q]||N_t[r]) = -t\, \mathbb{E}\left[\left(\nabla \log \frac{N_t[q]}{N_t[r]}\right)^2\right] < 0, \quad \text{if } q \neq r.$$

This is a standard fact but a proof is offered for the reader's convenience at the end of this section. Note that the Data Processing Inequality immediately implies that the KL divergence is non-increasing, but the claim is that it is actually decreasing.

Combining Lemma C.2 and Lemma C.3, we see that at high noise levels, distributions become *more compositional*, and thus better-approximated by local-conditional-scores. We now provide the proofs.

*Proof.* (Lemma C.2)

Define the "ideal" projective composition for any $\mathcal{J}$ by:

$$\mathcal{C}^\star_{\mathcal{J}}[p](x) := p(x_{M^{\complement}_{\mathcal{J}}}|\emptyset) \prod_{j\in\mathcal{J}} p(x_{M_j}|c_j).$$

By Lemma 3.3, we have that $s$ is exact for $\nabla \log \mathcal{C}^\star[p]$, and so $\widehat{p}(\cdot|c_{\mathcal{J}}) = c\mathcal{C}^\star_{\mathcal{J}}[p]$.

Thus we need to show that

$$KL(p(\cdot|c_{\mathcal{J}})||\mathcal{C}^\star_{\mathcal{J}}[p]) \leq \sum_{j\in\mathcal{J}} \varepsilon_j + \sum_{j\notin\mathcal{J}} \tilde{\varepsilon}_j + \varepsilon_b.$$

This is essentially a bound on a mean field approximation. First, note that for any $c$, we can rewrite $p(x|c)$ using the chain rule as

$$p(x|c) = p(x_{M_b}|c_{\mathcal{J}}) \prod_{j\in\mathcal{J}_{\text{all}}} p(x_{M_j}|c, x_{M_b}, x_{M_1}, \ldots, x_{M_{j-1}})$$

Then calculate

$$
\mathrm{KL}(p(\cdot|c_{\mathcal{J}})||\mathcal{C}_{\mathcal{J}}^{\star}[p]) \equiv \underset{p(x|c_{\mathcal{J}})}{\mathbb{E}} \left[ \log \frac{p(x|c_{\mathcal{J}})}{\mathcal{C}^{\star}[\vec{p}](x)} \right]
$$

$$
= \underset{p(x|c_{\mathcal{J}})}{\mathbb{E}} \left[ \log \frac{p(x_{M_b}|c_{\mathcal{J}}) \prod_j p(x_{M_j}|c_{\mathcal{J}}, x_{M_b}, x_{M_1}, \ldots, x_{M_{j-1}})}{p(x_{M_b}) \prod_{j \notin \mathcal{J}} p(x_{M_j}|\emptyset) \prod_{j \in \mathcal{J}} p(x_{M_j}|c_j)} \right]
$$

$$
= \sum_{j \in \mathcal{J}} \underset{p(x|c_{\mathcal{J}})}{\mathbb{E}} \left[ \log \frac{p(x_{M_j}|c_{\mathcal{J}}, x_{M_b}, x_{M_1}, \ldots, x_{M_{j-1}})}{p(x_{M_j}|c_j)} \right]
$$

$$
+ \sum_{j \notin \mathcal{J}} \underset{p(x|c_{\mathcal{J}})}{\mathbb{E}} \left[ \log \frac{p(x_{M_j}|c_{\mathcal{J}}, x_{M_b}, x_{M_1}, \ldots, x_{M_{j-1}})}{p(x_{M_j}|\emptyset)} \right]
$$

$$
+ \underset{p}{\mathbb{E}} \left[ \log \frac{p(x_{M_b}|c_{\mathcal{J}})}{p(x_{M_b})} \right]
$$

$$
= \sum_{j \in \mathcal{J}} \underset{p(x_{M_b}, x_{M_1}, \ldots, x_{M_{j-1}}|c_{\mathcal{J}})}{\mathbb{E}} \left[ \mathrm{KL}[p(x_{M_j}|c_{\mathcal{J}}, x_{M_b}, x_{M_1}, \ldots, x_{M_{j-1}})||p(x_{M_j}|c_j))] \right]
$$

$$
+ \sum_{j \notin \mathcal{J}} \underset{p(x_{M_b}, x_{M_1}, \ldots, x_{M_{j-1}}|c_{\mathcal{J}})}{\mathbb{E}} \left[ \mathrm{KL}[p(x_{M_j}|c_{\mathcal{J}}, x_{M_b}, x_{M_1}, \ldots, x_{M_{j-1}})||p(x_{M_j}|\emptyset))] \right]
$$

$$
+ \underset{p(x_{M_b^{\complement}}|c_{\mathcal{J}})}{\mathbb{E}} \left[ \mathrm{KL}[p(x_{M_b}|c_{\mathcal{J}})||p(x_{M_b}))] \right]
$$

$$
\leq \sum_{j \in \mathcal{J}} \sup_{x_{M_j^{\complement}}} \mathrm{KL}[p(x_{M_j}|c_{\mathcal{J}}, x_{M_j^{\complement}})||p(x_{M_j}|c_j)]
$$

$$
+ \sum_{j \notin \mathcal{J}} \sup_{x_{M_j^{\complement}}} \mathrm{KL}[p(x_{M_j}|c_{\mathcal{J}}, x_{M_j^{\complement}})||p(x_{M_j}|\emptyset)]
$$

$$
+ \sup_{x_{M_b^{\complement}}} \mathrm{KL}[p(x_{M_b}|c_{\mathcal{J}}, x_{M_b^{\complement}})||p(x_{M_b})]
$$

$$
\leq \sum_{j \in \mathcal{J}} \varepsilon_j + \sum_{j \notin \mathcal{J}} \tilde{\varepsilon}_j + \varepsilon_b
$$

$\square$

*Proof.* (Lemma C.3)

We want to show that

$$
t_2 \geq t_1 \implies \sup_y [KL(N_{t_2}[p](x)||N_{t_2}[p](x|y))] < \sup_y [KL(N_{t_1}[p](x)||N_{t_1}[p](x|y))]
$$

Let $t_2 \geq t_1$. By assumption, the supremum of $\sup_y [KL(N_{t_2}[p](x)||N_{t_2}[p](x|y))]$ is attained, so let $y_2^*$ be a value of $y$ that achieves the supremum for $t_2$. Then

$$
y_2^{\star} := argmax_y [KL(N_{t_2}[p](x)||N_{t_2}[p](x|y))]
$$

$$
\sup_y [KL(N_{t_1}[p](x)||N_{t_1}[p](x|y))] := KL(N_{t_2}[p](x)||N_{t_2}[p](x|y_2^{\star}))
$$

$$
\leq KL(N_{t_1}[p](x)||N_{t_1}[p](x|y_2^{\star})), \quad \text{by Lemma C.4}
$$

$$
\leq \sup_y [\mathrm{KL}(N_{t_1}[p](x)||N_{t_1}[p](x|y))]
$$

$\square$

*Proof.* (Lemma C.4) We want to show that

$$
\frac{\partial}{\partial t} \mathrm{KL}(N_t[q]||N_t[r]) = -t \, \mathbb{E} \left[ \left( \nabla \log \frac{N_t[q]}{N_t[r]} \right)^2 \right] < 0, \quad \text{if } q \neq r.
$$

This is a standard result but we provide a proof for the reader's convenience.

$$\mathrm{KL}(q||r) := \mathbb{E}_q[\log \frac{q}{r}] = \int q(x) \log \frac{q(x)}{r(x)} dx$$

$$N_t[p](x) := \int p(y) \mathcal{N}(x; y, t^2) dy$$

$$\mathbb{E}_{N_t[q]}[\log \frac{N_t[q]}{N_t[r]}] = \int N_t[q](x) \log \frac{N_t[q](x)}{N_t[r](x)} dx$$

$$\frac{\partial}{\partial t} \mathbb{E}_{N_t[q]}[\log \frac{N_t[q]}{N_t[r]}] = \frac{\partial}{\partial t} \int N_t[q](x) \log \frac{N_t[q](x)}{N_t[r](x)} dx$$

$$= \int \frac{\partial}{\partial t} N_t[q](x) \cdot \log \frac{N_t[q](x)}{N_t[r](x)} dx + \int N_t[q](x) \cdot \frac{\partial}{\partial t} \log \frac{N_t[q](x)}{N_t[r](x)} dx$$

$$\equiv I_1 + I_2$$

To work on integrals $I_1$, $I_2$, we use the following fact: $\frac{\partial}{\partial t} N_t[p](x) = t\nabla^2 N_t[p](x)$ (Equation (9)).

$$I1 \equiv \int \frac{\partial}{\partial t} N_t[q](x) \cdot \log \frac{N_t[q](x)}{N_t[r](x)} dx$$

$$= \int t\nabla^2 N_t[q](x) \cdot \log \frac{N_t[q](x)}{N_t[r](x)} dx, \quad \text{using } \frac{\partial}{\partial t} N_t[p] = t\nabla^2 N_t[p] \text{ as shown below}$$

$$= -t \int \nabla N_t[q](x) \cdot \nabla \log \frac{N_t[q](x)}{N_t[r](x)} dx, \quad \text{integration by parts}$$

$$= -t \int \nabla N_t[q](x) \cdot (\nabla \log N_t[q](x) - \nabla \log N_t[r](x)) \, dx,$$

$$= -t \int \nabla N_t[q] \cdot \left( \frac{\nabla N_t[q]}{N_t[q]} - \frac{\nabla N_t[r]}{N_t[r]} \right) dx$$

$$= t \int -\frac{\nabla N_t[q]^2}{N_t[q]} + \frac{\nabla N_t[r]\nabla N_t[q]}{N_t[r]} dx$$

$$I_2 \equiv \int N_t[q](x) \cdot \frac{\partial}{\partial t} \log \frac{N_t[q](x)}{N_t[r](x)} dx$$

$$\frac{\partial}{\partial t} \log \frac{N_t[q](x)}{N_t[r](x)} = \frac{\partial}{\partial t} \log N_t[q](x) - \frac{\partial}{\partial t} \log N_t[r](x)$$

$$= \frac{1}{N_t[q](x)} \frac{\partial}{\partial t} N_t[q](x) - \frac{1}{N_t[r](x)} \frac{\partial}{\partial t} N_t[r](x)$$

$$= \frac{t}{N_t[q](x)} \nabla^2 N_t[q](x) - \frac{t}{N_t[r](x)} \nabla^2 N_t[r](x)$$

$$\implies I_2 = t \int N_t[q](x) \cdot \left( \frac{\nabla^2 N_t[q](x)}{N_t[q](x)} - \frac{\nabla^2 N_t[r](x)}{N_t[r](x)} \right) dx$$

$$= -t \int \frac{N_t[q](x)}{N_t[r](x)} \nabla^2 N_t[r] dx, \quad \text{since } \int \nabla^2 N_t[q] dx = \nabla N_t[q]|_{-\infty}^{\infty} = 0$$

$$= t \int \nabla \left( \frac{N_t[q](x)}{N_t[r](x)} \right) \nabla N_t[r] dx, \quad \text{integration by parts}$$

$$= t \int \frac{\nabla N_t[r]\nabla N_t[q]}{N_t[r]} - \frac{\nabla N_t[r]^2 N_t[q]}{N_t[r]^2} dx$$

Therefore

$$
\begin{aligned}
I_1 + I_2 &= t\int -\frac{\nabla N_t[q]^2}{N_t[q]} + \frac{\nabla N_t[r]\nabla N_t[q]}{N_t[r]}dx + t\int \frac{\nabla N_t[r]\nabla N_t[q]}{N_t[r]} - \frac{\nabla N_t[r]^2 N_t[q]}{N_t[r]^2}dx\\
&= -t\int \frac{\nabla N_t[q]^2}{N_t[q]} - 2\frac{\nabla N_t[r]\nabla N_t[q]}{N_t[r]} + \frac{\nabla N_t[r]^2 N_t[q]}{N_t[r]^2}dx\\
&= -t\int N_t[q]\left(\frac{\nabla N_t[q]}{N_t[q]} - \frac{\nabla N_t[r]}{N_t[r]}\right)^2 dx\\
&= -t\,\mathbb{E}\left[\left(\nabla \log \frac{N_t[q]}{N_t[r]}\right)^2\right]
\end{aligned}
$$

This concludes the proof. Note that the fact used earlier can be shown as follows:

$$\text{Claim:}\quad \frac{\partial}{\partial t}N_t[p](x) = t\nabla^2 N_t[p](x) \tag{9}$$

To see this:

$$
N_t[p](x) := \int p(y)\phi(y)dy, \quad \phi(y;x,t) := \frac{1}{\sqrt{2\pi t^2}}e^{-\frac{(x-y)^2}{2t^2}}
$$

$$
\frac{\partial}{\partial t}\phi(y;x,t) = \frac{1}{\sqrt{2\pi}}\left[-\frac{1}{t^2} + \frac{(x-y)^2}{t^4}\right]e^{-\frac{(x-y)^2}{2t^2}} = \left[\frac{(x-y)^2}{t^3} - \frac{1}{t}\right]\phi(y;x,t)
$$

$$
\frac{\partial^2}{\partial x^2}\phi(y;x,t) = \frac{1}{\sqrt{2\pi t^2}}\left[\frac{(x-y)^2 - t^2}{t^4}\right]e^{-\frac{(x-y)^2}{2t^2}} = \frac{1}{t}\left[\frac{(x-y)^2}{t^3} - \frac{1}{t}\right]\phi(y;x,t)
$$

$$
\implies \frac{\partial}{\partial t}N_t[p](x) = \int p(y)\frac{\partial}{\partial t}\phi(y;x,t)dy = \int p(y)\left[\frac{(x-y)^2}{t^3} - \frac{1}{t}\right]\phi(y;x,t)dy
$$

$$
\nabla^2 N_t[p](x) = \int p(y)\frac{\partial^2}{\partial x^2}\phi(y;x,t)dy = \int p(y)\frac{1}{t}\left[\frac{(x-y)^2}{t^3} - \frac{1}{t}\right]\phi(y;x,t)dy
$$

$$
\implies t\nabla^2 N_t[p](x) = \frac{\partial}{\partial t}N_t[p](x)
$$

$\square$

## D. Feature-space theory

In this section we discuss the relationship between CPC and LCS in feature space. Inspired by the feature-space adaptation of PC in (Bradley et al., 2025), we define feature-space conditional projective composition as follows:

**Definition D.1** (Feature-space Conditional Projective Composition (F-CPC)). We say that $p(x|c)$ is a F-CPC under an invertible transform $\mathcal{A} : \mathbb{R}^n \to \mathbb{R}^n$ (mapping pixel-space to feature-space), if $\mathcal{A}\sharp p$ (where $\sharp$ denotes the pushforward) is a CPC according to Definition 3.2, that is: there exist disjoint sets $M_j$ for all conditions $j \in \mathcal{J}_{\text{all}}$ such that, for any set of conditions $\mathcal{J} \in \mathcal{J}_{\text{all}}$,

$$(\mathcal{A}\sharp p)(z|c_{\mathcal{J}}) := (\mathcal{A}\sharp p)(z_{M_{\mathcal{J}}^{\complement}}|\emptyset)\prod_{j\in\mathcal{J}}(\mathcal{A}\sharp p)(z_{M_j}|c_j), \quad \text{where } z := \mathcal{A}(x). \tag{10}$$

That is, $p$ is an F-CPC if it has CPC compositional structure under an appropriate feature-space mapping. In order to exploit this sparse dependency structure, the model now needs to learn the associated feature-space transform and its inverse, in addition to the local subsets $N_i, L_i$. For F-CPC distributions, Corollary 3.4 follows directly from Lemma 3.3. We formally restate and prove Corollary 3.4:

*Corollary 3.4* (LCS is exact for CPC in feature-space; formal). Suppose that $p(x|c)$ is an F-CPC (Definition D.1) under an invertible transform $\mathcal{A} : \mathbb{R}^n \to \mathbb{R}^n$, with subsets $\{M_j : j \in \mathcal{J}_{\text{all}}\}$. Letting $z := \mathcal{A}(x)$ consider the specific local-conditional score $s$ given by

$$s_{\mathcal{A}}^t[z|c_{\mathcal{J}}](i) := \nabla_z \log(\mathcal{A}\sharp p)^t[z_{N_i^t}|c_{L_i^t(\mathcal{J})}](i),$$

where $N_i, L_i$ are defined as in Lemma 3.3 w.r.t. the subsets $\{M_j\}$. Then $s_{\mathcal{A}}^t[z|c_{\mathcal{J}}]$ is exact for the score of $\mathcal{A}\sharp p(z|c_{\mathcal{J}})$ w.r.t. $z$:

$$s_{\mathcal{A}}^t(z|c_{\mathcal{J}}) = \nabla_z \log(\mathcal{A}\sharp p)^t(z|c_{\mathcal{J}}), \quad \forall\mathcal{J}.$$

*Proof.* Apply Lemma 3.3 to $\mathcal{A}\sharp p(z|c)$. □

Sampling with this local score is more complex than it looks, however! As discussed in (Bradley et al., 2025), since the noising operator does not commute with $\mathcal{A}$, that is, $(\mathcal{A}\sharp p)^t \neq \mathcal{A}\sharp(p^t)$, sampling from $p$ using $s_{\mathcal{A}}^t$ would actually requires the process

$$N_t[p] \xrightarrow{\mathcal{A}^t} N_t[\mathcal{A}\sharp p_i] \xrightarrow{s_{\mathcal{A}}^t} N_{t-1}[\mathcal{A}\sharp p] \xrightarrow{(\mathcal{A}^{t-1})^{-1}} N_{t-1}[p] \to \ldots \to p,$$

where $N_t$ denotes the noising operator, i.e. $N_t[p] := p^t$, and $\mathcal{A}^t$ corrects for the non-commutativity:

$$\mathcal{A}^t \sharp N_t[p] := N_t[\mathcal{A}\sharp p].$$

Therefore, it is not enough to learn a single transform $\mathcal{A}$ and its inverse $\mathcal{A}^{-1}$ – the network actually needs to learn a time-dependent transform/inverse pair $\mathcal{A}^t$, $\mathcal{A}^{-t}$ accounting for the interaction between $\mathcal{A}$ and the noising process at each time $t$, which actually depends on the (unknown) distribution $p$. How feasible this is is currently unclear.

### D.1. F-LCS Heuristic

Finally, we prove Lemma 3.5, which provides a necessary-but-not-sufficient condition for F-LCS.

*Proof.* (Lemma 3.5) First, note that for scores of any distribution $p$, and any fixed choice of $x$,

$$s^t(x|c_i)[k] = \begin{cases} \nabla \log p^t(x_{M_i}|c_i)[k], & \forall k \in M_i \\ \nabla \log p^t(x_{M_\ell}|\emptyset)[k], & \forall k \in M_\ell, \quad \ell \neq i \text{ (including } \ell = b) \end{cases}$$

$$d_i^t(x)[k] := s^t(x|c_i) - s^t(x|\emptyset)$$

$$= \begin{cases} \nabla \log \frac{p^t(x_{M_i}|c_i)[k]}{p^t(x_{M_i}|\emptyset)[k]}, & \forall k \in M_i \\ 0, & \forall k \notin M_i \end{cases}$$

$$\implies d_i^t(x)^T d_j^t(x) = 0, \quad \forall i \neq j, \quad \text{since } M_i \cap M_j = \emptyset,$$

where in the second-to-last line we used the fact that the gradient of a function depending only on a subset of variables has zero entries in the coordinates outside that subset. The same orthogonality result also holds for $x_0$-parametrized networks since the score is related to the conditional mean by $\nabla \log p^t(x_t) := \frac{1}{\sigma_t^2} \mathbb{E}[x_0 - x_t | x_t]$, therefore $d_i^t(x) \propto \mathbb{E}_{p(x_0|x_t,c_i)}[x_0|x_t] - \mathbb{E}_{p(x_0|x_t,\emptyset)}[x_0|x_t]$.

Similarly, we can take an expectation over an arbitrary distribution $x \sim q$ and obtain the following orthogonality result:

$$\mathop{\mathbb{E}}_{x \sim q}[s^t(x|c_i)][k] = \begin{cases} \mathbb{E}_{x \sim q}[\nabla \log p^t(x_{M_i}|c_i)][k], & \forall k \in M_i \\ \mathbb{E}_{x \sim q}[\nabla \log p^t(x_{M_\ell}|\emptyset)][k], & \forall k \in M_\ell, \quad \ell \neq i \text{ (including } \ell = b) \end{cases}$$

$$d_i^t(q)[k] := \mathop{\mathbb{E}}_{x \sim q}[s^t(x|c_i)][k] - \mathop{\mathbb{E}}_{x \sim q}[s^t(x|\emptyset)][k] = 0, \quad \forall k \notin M_i$$

$$\implies d_i^t(q)^T d_j^t(q) = 0, \quad \forall i \neq j, \quad \text{since } M_i \cap M_j = \emptyset.$$

Applying the result to F-LCS scores $s_{\mathcal{A}}^t(z|c) := \nabla_z \log(\mathcal{A}\sharp p)^t(z|c)$, with $q \sim \mathcal{A}\sharp \mathcal{N}(0, \sigma_{t_{\max}})$, at time $t = t_{\max}$, which corresponds to evaluating the score in feature-space at the first denoising step (when the input is Gaussian noise) – gives

$$d_i := \mathop{\mathbb{E}}_{\eta_{\mathcal{A}}}[s_{\mathcal{A}}^{t_{\max}}(\eta_{\mathcal{A}}|c_i)] - \mathop{\mathbb{E}}_{\eta_{\mathcal{A}}}[s_{\mathcal{A}}^{t_{\max}}(\eta_{\mathcal{A}}|\emptyset)], \quad \eta_{\mathcal{A}} \sim \mathcal{A}\sharp \mathcal{N}(0, \sigma_{t_{\max}})$$

$$\implies d_i^T d_j = 0, \quad \forall i \neq j.$$

□

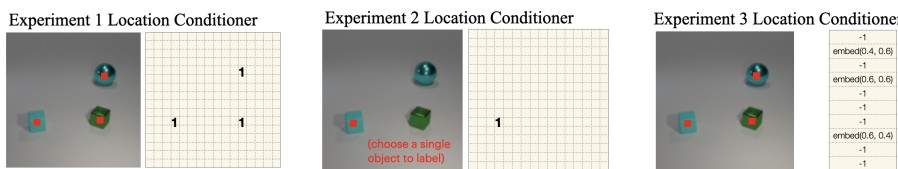

*Figure 7.* Examples of location conditioning used in Experiments 1, 2, 3 in Figure 1.

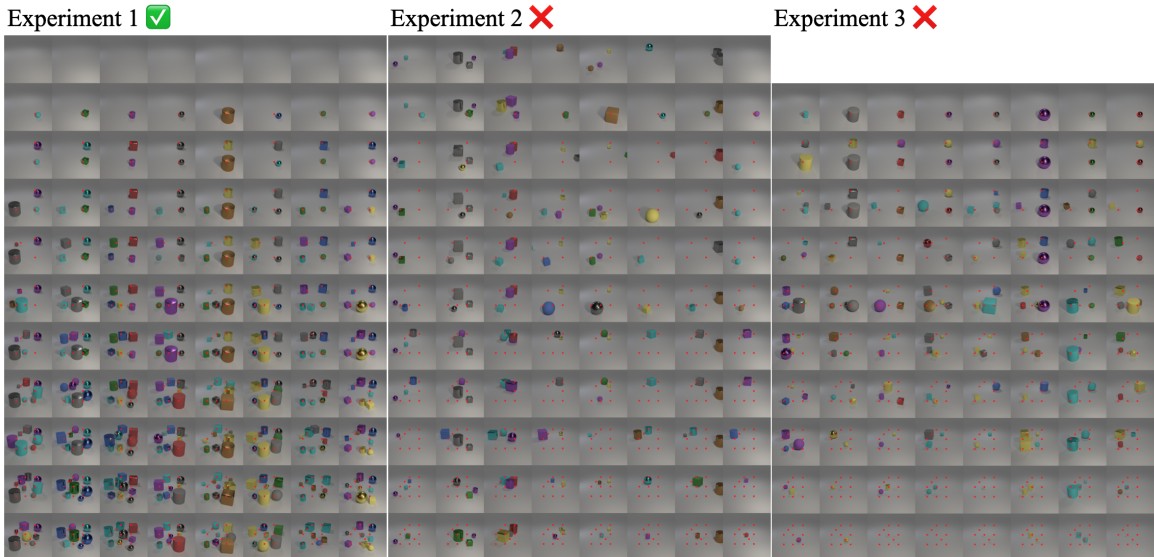

*Figure 8.* **Length-generalization in Experiment 1, 2, 3 on** $1 - M$ **objects**. We tested length-generalization from $K = 0$ to 10 conditioned locations in each model (each row shows 8 samples for a particular $K$). (Note Exp. 3 does not support $K = 0$.)

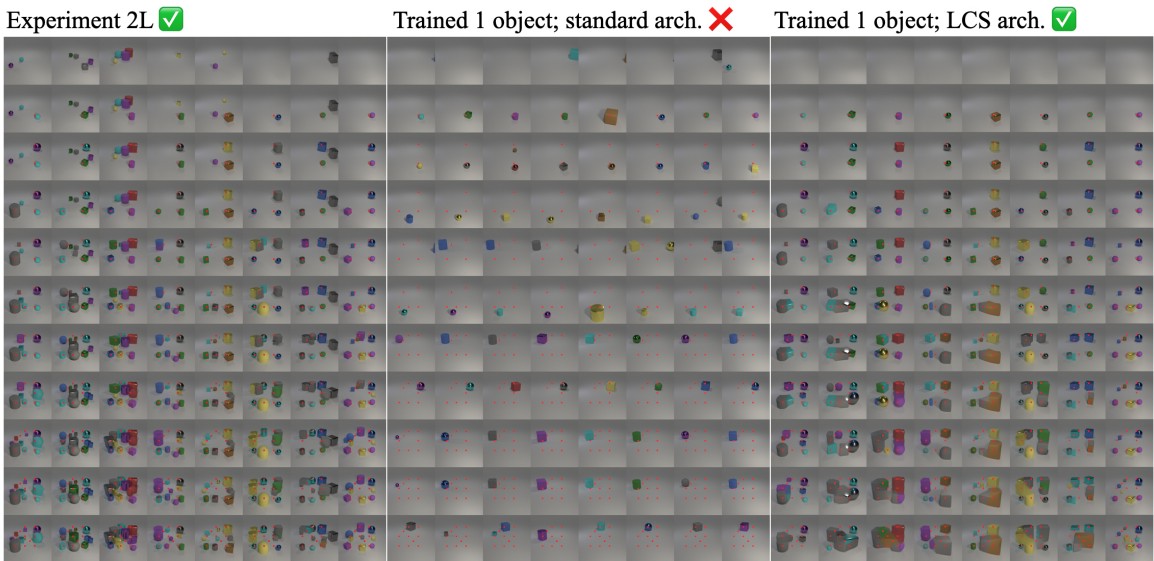

*Figure 9.* **Local causal intervention enables length generalization.** (Left) Additional samples from Exp. 2L of Figure 1 (see also Section E.3). (Center and Right) Length-generalization in two different location-conditioned models, both trained on images with only a *single* object (and conditioned on its single location). (Center) A model with the standard EDM2 architecture does not length-generalize: it always generates exactly one object (even when conditioned on zero locations). (Right) A model with an explicitly-enforced local architecture as in Exp. 2L length-generalizes up to 5 objects, albeit with some artifacts (objects "merging" into each other). Although it does not perform as well as the Exp. 2L local model trained on 1-3 objects, any length-generalization after training on only one object is remarkable. (In Section E.2, we hypothesize that training on more objects, e.g. 1-3, may improve length-generalization by allowing the model to learn *clusters* of objects).

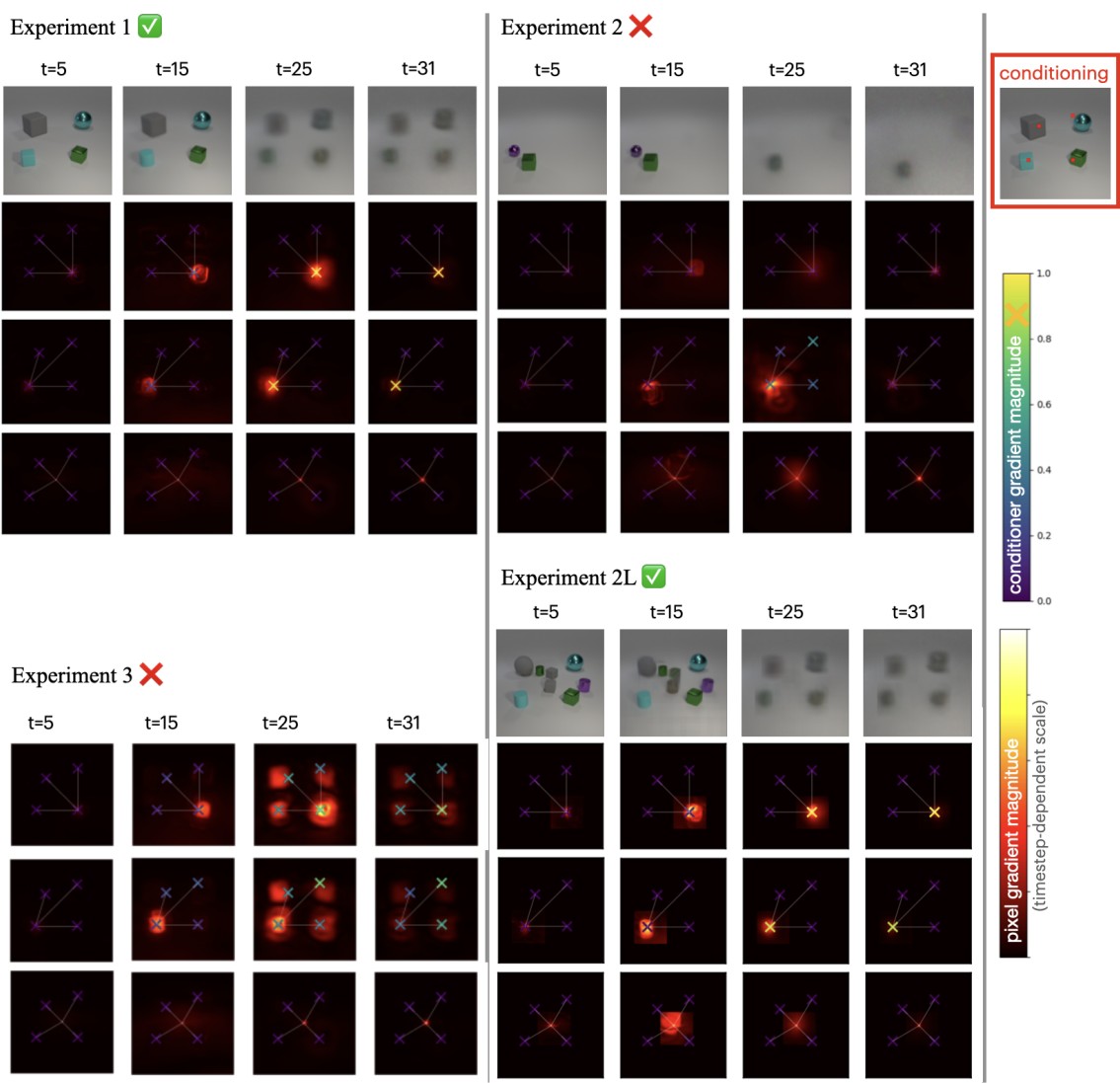

*Figure 10.* Additional detail for Figure 2. Locality structures in location-conditioned CLEVR models (Exps. 1, 2, 3, 2L of Figure 1). All models are conditioned on 4 locations (OOD). Each column represents a timestep $t$. Top row shows the predicted denoised images via learned scores. Lower rows (evaluated at two conditioned and one unconditioned location) show heatmaps of the pixel gradient magnitude (average absolute of Jacobian from one pixel to all other pixels), and the conditional gradient magnitude marked with × (with the "gradient" estimated via a finite difference of the score computed with and without each conditioner).

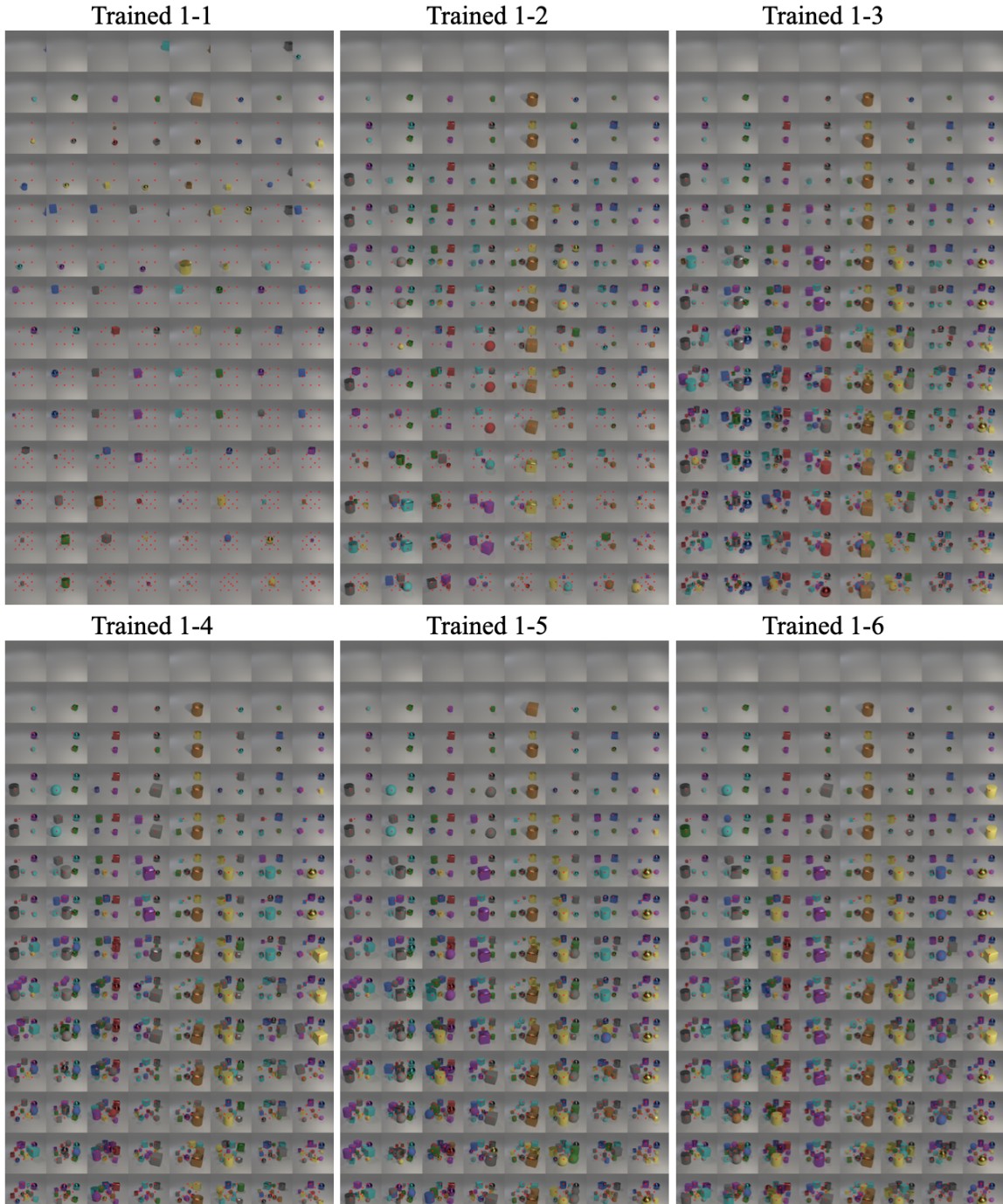

*Figure 11.* **Length-generalization in Experiment 1 model trained on** $1 - M$ **objects**. We tested length-generalization from $K = 0$ to $K = 12$ conditioned locations in each model (each row shows 8 samples for a particular $K$).

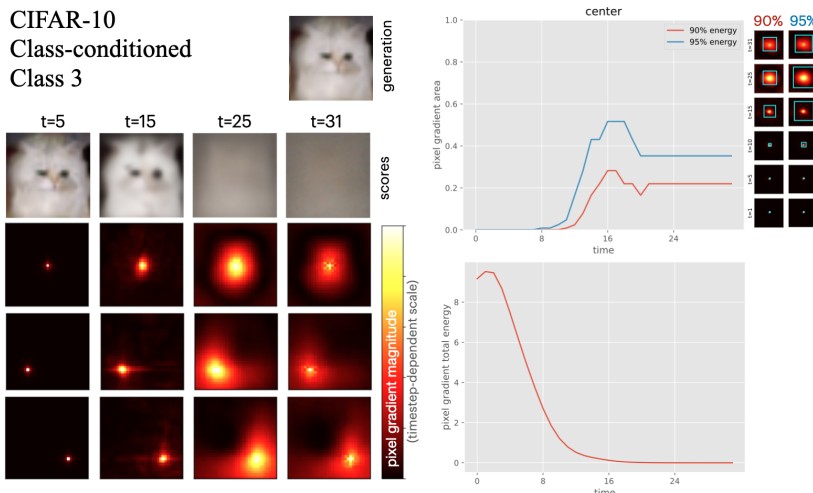

*Figure 12.* **Pixel locality structure across time for CIFAR-10**, using the same EDM2 architecture from the CLEVR experiments. (Left) Each column represents a timestep $t$, and the top row shows the predicted denoised images via learned scores. Lower rows show heatmaps of the pixel gradient magnitude (average absolute Jacobian from an selected pixel to all other pixels). Evaluation at three pixel locations confirms prior empirical observations of (Kamb & Ganguli, 2024; Niedoba et al., 2024) that the effective local neighborhood size is large at high noise levels. (Right) Locality metrics as in Figure 13; details in Section F.

# E. CLEVR: details and additional experiments

## E.1. CLEVR Dataset, architecture, and training details

We used the CLEVR (Johnson et al., 2017) dataset generation procedure[3] to generate custom datasets with the default objects, shapes, sizes, colors, but different counts. We generated various datasets with 1 to $K$ objects, with 500,000 samples for each object count, for $K = 1, \ldots, 6$ – for example, models trained on 1-3 objects saw a total of 1,500,000 samples. The image resolution is $128 \times 128$. Note that objects can interact with each other in this dataset: potentially occluding or casting shadows on each other.

Our experiments cover a few different conditioning setups. Grid-style location-conditioning conditions on 2D object locations, implemented as an 2D integer array representing a `grid_size × grid_size` grid over the image recording the count of objects whose center falls within the grid cell, as shown in Figure 7. The count is usually 0 or 1 but can be greater than 1 if object centers happen to land within the same grid cell. We take `grid_size=16` in all experiments. We either record the locations of all objects, or just one of the objects (randomly selected), in the conditioning grid, depending on the experiment.

List-style location conditioning also conditions on 2D object locations, but lists the (embedded) xy-locations of each object in an array padded with enough slots for up to 10 objects (with each location placed in a randomly chosen slot).

Color conditioning is implemented as a 8-dimensional integer array (there are 8 possible colors) indicating the count of objects with the corresponding color. In all experiments we condition only a single attribute (either location or color) at a time, with all other attributes sampled randomly and not conditioned on.

The 12 locations used for the location-conditioned CLEVR experiments are

```
locations = ([[0.65, 0.65], [0.65, 0.25], [0.25, 0.65], [0.35, 0.35],
[0.45, 0.65], [0.45, 0.25], [0.65, 0.45], [0.25, 0.45],
[0.45, 0.45], [0.55, 0.55], [0.35, 0.55], [0.55, 0.35]]),
```

and the colors are

```
colors = ([blue, brown, cyan, gray, green, purple, red, yellow]).
```

We used our own functionally equivalent re-implementation of the EDM2 (Karras et al., 2024) U-net architecture. We used

---

[3]https://github.com/facebookresearch/clevr-dataset-gen

| Experiment | $K_{\max}$ | Automated classifier counts | | | | | | | | | | |
|---|---|---|---|---|---|---|---|---|---|---|---|---|
| | | 0 | 1 | 2 | 3 | 4 | 5 | 6 | 7 | 8 | 9 | 10 |
| Exp.1, trained 1 | 1 | 0 | 1021 | 3 | 0 | 0 | 0 | 0 | 0 | 0 | 0 | 0 |
| Exp.1, trained 1-3 | 7 | 0 | 0 | 0 | 0 | 0 | 3 | 205 | 808 | 8 | 0 | 0 |
| Exp.1, trained 1-5 | 10 | 0 | 0 | 0 | 0 | 0 | 0 | 0 | 0 | 8 | 327 | 689 |
| Exp.2, trained 1 | 1 | 0 | 1021 | 3 | 0 | 0 | 0 | 0 | 0 | 0 | 0 | 0 |
| Exp.2, trained 1-3 | 2 | 0 | 342 | 387 | 293 | 2 | 0 | 0 | 0 | 0 | 0 | 0 |
| Exp.2, trained 1-5 | 3 | 0 | 175 | 267 | 208 | 185 | 168 | 21 | 0 | 0 | 0 | 0 |
| Exp.3, trained 1-3 | 3 | 0 | 63 | 170 | 790 | 1 | 0 | 0 | 0 | 0 | 0 | 0 |
| Exp.3, trained 1-5 | 6 | 0 | 0 | 0 | 0 | 6 | 606 | 367 | 45 | 0 | 0 | 0 |
| Exp.2L, trained 1 | 5 | 0 | 0 | 0 | 0 | 262 | 653 | 107 | 2 | 0 | 0 | 0 |
| Exp.2L, trained 1-3 | 9 | 0 | 0 | 0 | 0 | 0 | 0 | 2 | 63 | 291 | 351 | 317 |
| Exp.2L, trained 1-5 | 10 | 0 | 0 | 0 | 0 | 0 | 0 | 2 | 11 | 66 | 189 | 756 |
| Exp.3L, trained 1-3 | 9 | 0 | 0 | 0 | 0 | 0 | 15 | 64 | 187 | 316 | 228 | 214 |

*Table 2.* $K_{\max}$ counts for Table 1 experiments, evaluated using the automated classifier over 1024 samples. The "Len" column indicates the number of locations conditioned on (equal to $K_{\max}$); columns represent number of objects generated at conditioned locations, and rows contain counts of images that contain the number of objects listed in the column.

the smallest model architecture, e.g. `edm2-img64-xs` from `https://github.com/NVlabs/edm2`. This model has a base channel width of 128, resulting in a total of 124M trainable weights.

In all experiments, the model is trained with a batch size of 2048 over $128 \times 2^{20}$ samples, repeating samples if needed. Our training procedure is identical to EDM2 (Karras et al., 2024) except that we do weight renormalization after the weights are updated. At inference, we use raw conditional diffusion scores, without applying any guidance/CFG (Ho & Salimans, 2022).

### E.2. Length generalization evaluation

In Table 1, we define $K_{\max}$ as the maximum value such that the model "sometimes succeeds" for every $1 \le K \le K_{\max}$, with "success" defined as generating $K$ objects at least 50% of the time, and at least $K - 2$ objects at least 90% of the time. This metric is intentionally generous as it is intended to capture the largest $K$ for which the model "sometimes succeeds", rather than requiring perfect performance. To assess these criteria in Table 1, we use a ResNet-18 classifier (99.74% test accuracy) trained to count 1-10 CLEVR objects at specified locations, evaluated over 1024 samples per configuration. Note that the classifier counts the total number of objects, so for models that draw "extra" objects at non-conditioned locations (e.g. Exp. 2, which was trained with only one location labeled but draws additional objects), $K_{\max}$ may slightly overestimate the number of objects at *correct* conditioned locations. Nevertheless, this does not materially affect the conclusions: Exp. 2 still fails to consistently place objects at conditioned locations beyond ∼3, even when trained on 1-5 objects. We test on up to $K = 12$ locations. In Table 2 we provide the complete counts for all experiments shown in Table 1. Table 3 provides a similar analysis for additional experiments with other conditioning configurations such as labeling a random number of objects.

In Table 1 and Figure 11 we observe improvements in length-generalization as we increase the maximum number $M$ of objects the model was trained on, for models that length-generalize at all (i.e. Experiments 1 and 2L). We note that for larger numbers of locations $K$ at inference-time, the objects become crowded (less independent). We hypothesize that in general, models trained on $1 - M$ objects could learn to represent clusters of $1 - M$ objects, as well as how to compose multiple clusters. (This is still consistent with our theory: it is a conditional projective composition where the conditioners are *subsets*. If the underlying data has this type of compositional structure, a trained model could learn to group individual conditioners into subsets in order to exploit it.) If this is the case, a model trained on 1-3 objects could generate, for instance, 12 objects, by composing 4 clusters of 3 objects each. This would mean that models trained on more objects (larger $M$) could more easily length-generalize to higher $K$ where objects become crowded.

#### E.2.1. FIGURE 3 DETAILS

In Figure 3 plots conditional locality vs. length generalization for a variety of models. Each model is shown in a different color (with different shapes indicating different stages during training). The early, mid, and late stages are 16777216, 33554432, 134217728 images seen, respectively. The color assignments, conditioning variants, and $K_{\max}$ values for each

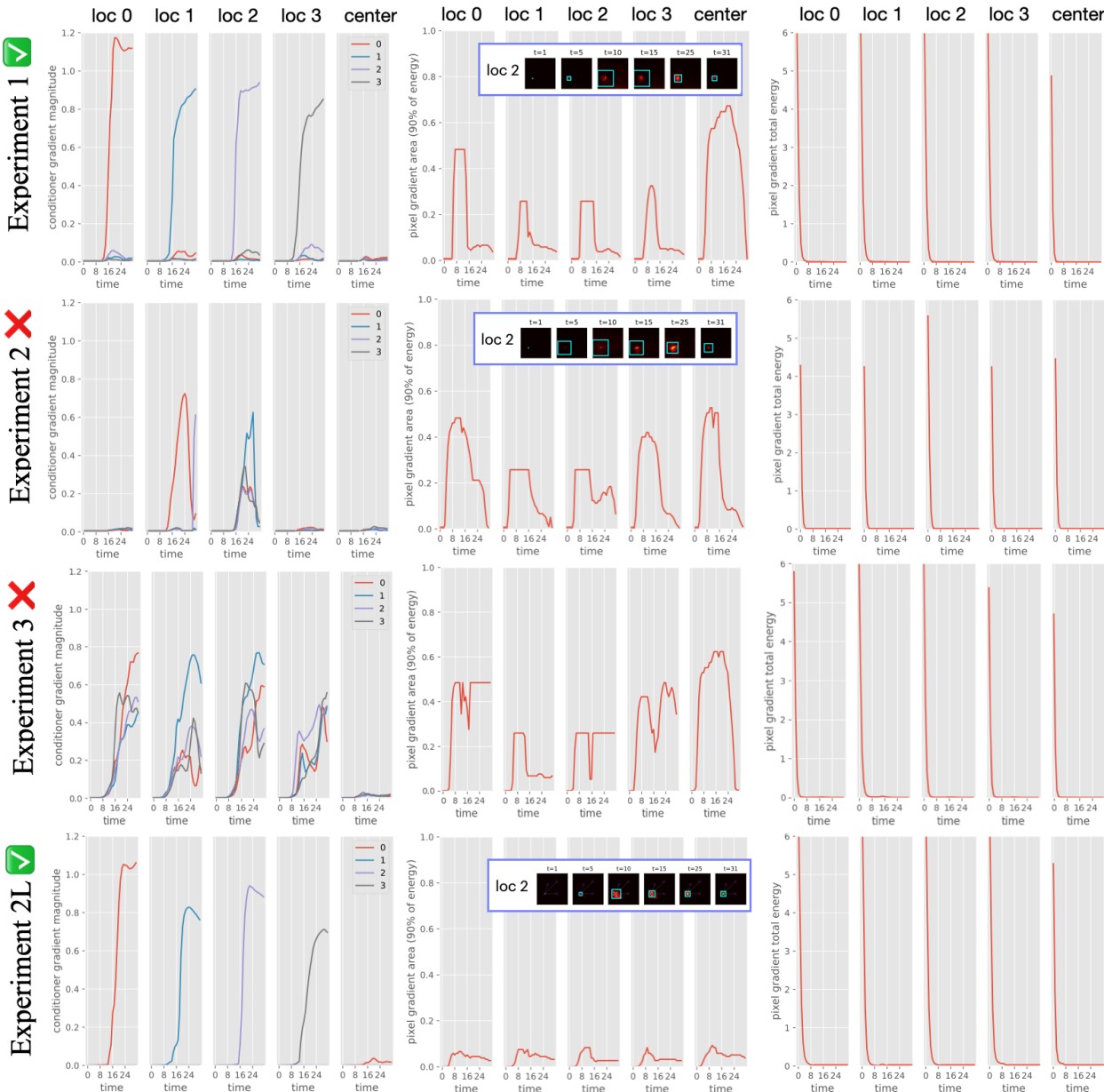

*Figure 13.* **Locality metrics for CLEVR length generalization** for Experiments 1, 2, 3, and 2L, conditioned OOD on 4 locations, as in Figure 2. (Left) Conditioner gradient magnitudes at selected pixels (loc 0–3 and center), with colors indicating individual conditioners. Experiment 1 shows strong conditional-locality at high noise (each pixel responds only to its corresponding conditioner), whereas Experiment 2 and 3 exhibit non-local responses. Experiment 2L has conditional-locality explicitly enforced by the architecture. (Middle) Pixel gradient area required to cover 90% of score gradient "energy" (sum of squared magnitudes). Insets illustrate the selected square regions at loc 2 at a few timesteps. We observe similar pixel-locality between Experiments 1 and 2; the pixel gradients are highly localized at both high and low noise, but delocalize during intermediate timesteps. Experiment 3 exhibits pixel non-locality even at high noise. Experiment 2L has pixel-locality explicitly enforced by the architecture. (Right) Total pixel gradient energy (over entire image), which is higher at low noise levels, consistent with conditioners dominating the score at high noise and pixel interactions emerging later in denoising.

*Table 3*. **Additional location-conditioning experiments** (Figure 3). $K_{max}$ values and count distributions for additional conditioning variants beyond Table 1: *All labels* means every object was labeled (as in Exp. 1), *Single label* means only one object (randomly selected) was labeled (as in Exp. 2), and *Rand num labels* means a random number of objects were labeled. The "Color" column indicates the color used in Figure 3. $K_{max}$ is evaluated using the automated classifier over 1024 samples at the final training epoch.

| Experiment | Color | $K_{max}$ | Automated classifier counts | | | | | | | | | | |
|---|---|---|---|---|---|---|---|---|---|---|---|---|---|
| | | | 0 | 1 | 2 | 3 | 4 | 5 | 6 | 7 | 8 | 9 | 10 |
| All labels, trained 1 | brown | 1 | 0 | 1021 | 3 | 0 | 0 | 0 | 0 | 0 | 0 | 0 | 0 |
| All labels, trained 1-2 | orange | 5 | 0 | 0 | 0 | 69 | 373 | 580 | 2 | 0 | 0 | 0 | 0 |
| All labels, trained 1-3 | red | 7 | 0 | 0 | 0 | 0 | 0 | 3 | 205 | 808 | 8 | 0 | 0 |
| All labels, trained 1-5 | blue | 10 | 0 | 0 | 0 | 0 | 0 | 0 | 0 | 0 | 8 | 327 | 689 |
| Single label, trained 1-3 | cyan | 2 | 0 | 342 | 387 | 293 | 2 | 0 | 0 | 0 | 0 | 0 | 0 |
| Rand labels, trained 1-2 | purple | 3 | 0 | 0 | 13 | 1009 | 2 | 0 | 0 | 0 | 0 | 0 | 0 |
| Rand labels, trained 1-3 | green | 5 | 0 | 0 | 0 | 0 | 151 | 873 | 0 | 0 | 0 | 0 | 0 |

experiment are given in Table 3.

In Figure 3 (Right), the $x$-axis shows length-generalization, defined as the number of locations to which the model can generalize *beyond* the number on which it was trained (e.g. +4 for a model trained on 1-3 locations that generalizes to 7). The number of locations to which the model can generalize is evaluated via $K_{max}$ as described in Section E.2. The complete counts are given in Table 3. The conditional locality ($y$-axis) metric is described in Section F.

### E.3. Details of Experiment 2L: Local Patch-Based Architecture Intervention.

For Experiment 2L (Figure 1 and 9), we developed a local variant of the EDM2 model that processes images as a grid of overlapping patches. The image is divided into a `grid_size` × `grid_size` grid of cells, where each cell has size `m = resolution / grid_size`. We set `grid_size = 16` to match the location-conditioning grid. For each grid cell at position $(i, j)$, we extract a patch of size $M = (2k + 1) \times m$, where $k$ is the neighborhood radius. We used $k = 2$ for the experiments in this paper. Each patch is conditioned only on the location-conditioners that fall within the patch – that is, a $(2k + 1) \times (2k + 1)$ subgrid of the full conditioning grid. Each patch is also conditioned on its absolute location (that is, the model is not equivariant). This is implemented by appending the patch center coordinates $i, j$ to the flattened location-conditioner grid to form the complete conditioner. The training procedure uses a patch sampling approach balancing positive and negative examples for efficiency. For each training image, we randomly sample two patches: one "positive" patch with at least one active conditioner, and one "negative" patch with no active conditioners. Standard EDM2 loss is applied to each patch and losses are averaged.

At inference, we reconstruct the full image by processing each grid cell as follows: extract the corresponding noisy patch and local conditioner; denoise the patch using the trained local model; copy the center region (the single cell) of the denoised patch back to the full image.

In Experiment 2L, we trained the local model on the same dataset and location-conditioning (labeling only a single object) as in Exp. 2, showing that Exp. 2L length-generalizes while Exp. 2 fails. As a test, we also verified that setting $k = 8$ (which for grid size 16 makes the patches the size of the image) reproduces the behavior of Exp. 2 (i.e. length-generalization fails). We also trained the local model on a dataset with only a single object per image and show that it length-generalizes up to 5 objects in this case, whereas a standard model trained on only one object per image only generates one object per image at test time, regardless of conditioning (Figure 9).

### E.4. Experiment 3L

Analogous to Experiment 2L, we apply a local causal intervention to the non-length-generalizing Experiment 3 and show that it enables length-generalization, as shown in Figure 14. In Experiment 3L, the model denoises individual patches conditioned only on the location-conditioners that fall within the patch, with conditioners represented as a list as in Experiment 3. The setup is as described in Section E.3, with the only difference being the list-style conditioner. Specifically, the original conditioner lists the locations of each object in an array padded with enough slots for up to 10 objects (with each object placed in a randomly-chosen slot); the patch conditioner includes only the conditioned locations that fall within the current patch, re-centered relative to the patch center, and placed within the padded array in their original random slots. Each patch

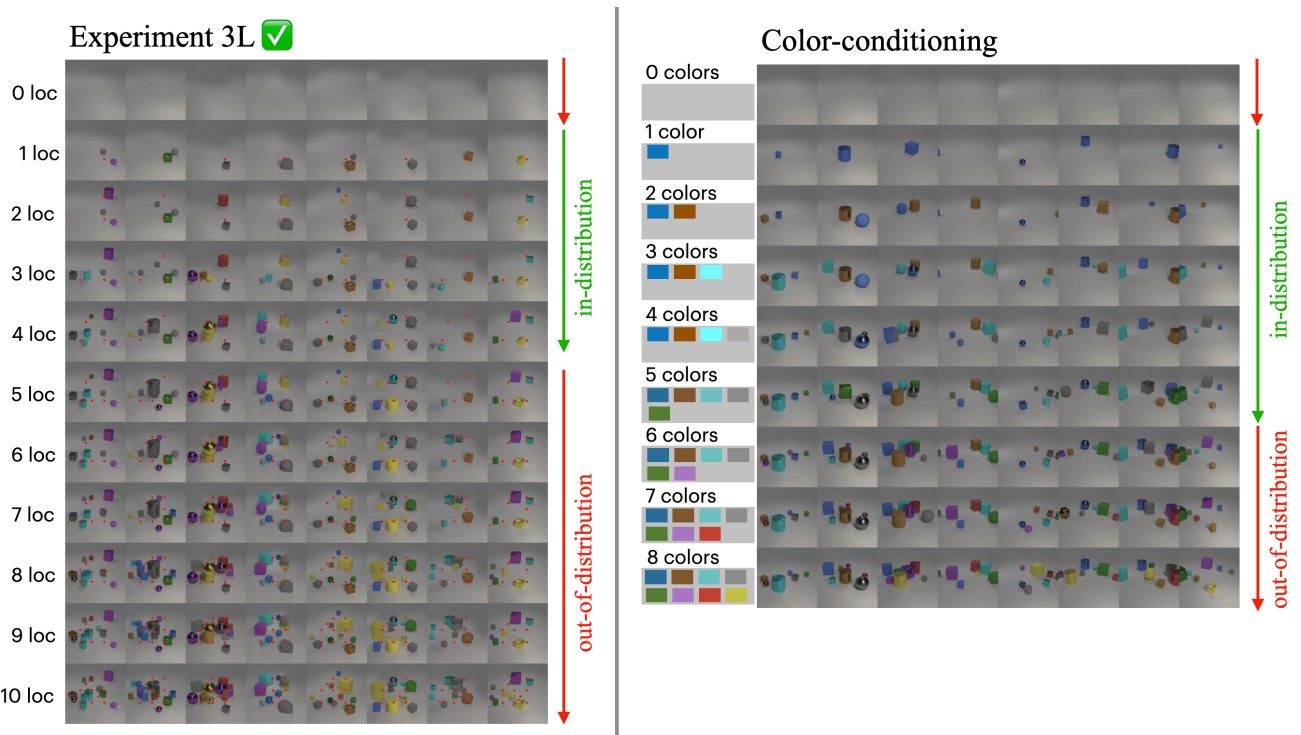

*Figure 14.* (Left) **Experiment 3L** applies a causal intervention to the failing Exp. 3: we modify the model architecture to explicitly enforce local conditional scores, use the same training data (1-3 objects) and conditioning as Exp. 3, and find that Exp. 3L length-generalizes while Exp. 3 failed. Note that both Exp. 3 and Exp. 3L struggle somewhat with location-accuracy even in-distribution, and this inaccuracy persists out-of-distribution for Exp. 3L. Nevertheless, Exp. 3L displays much stronger length-generalization than Exp. 3, enabling much higher object counts. (Right) **Length generalization in color-conditioned CLEVR.** We test length generalization in a CLEVR model conditioned on *colors* (rather than locations), trained on 1-5 objects. Each row is conditioned on the first $K$ colors in order (blue, brown, cyan, gray, green, purple, red, yellow), e.g. row 4 is conditioned on blue, brown, cyan, gray. We test up to 8 colors ($> 5$ colors is OOD), and find generalization up to 7 colors, suggesting that compositional structure may exist in a learned feature-space.

| Experiment | $K_{\max}$ | Manual counts | | | | | | | | | | |
|---|---|---|---|---|---|---|---|---|---|---|---|---|
| | | 0 | 1 | 2 | 3 | 4 | 5 | 6 | 7 | 8 | 9 | 10 |
| Color, trained 1 | 1 | 0 | 64 | 0 | 0 | 0 | 0 | 0 | 0 | 0 | 0 | 0 |
| Color, trained 1-3 | 4 | 0 | 0 | 2 | 29 | 33 | 0 | 0 | 0 | 0 | 0 | 0 |
| Color, trained 1-5 | 7 | 0 | 0 | 0 | 0 | 0 | 6 | 42 | 16 | 0 | 0 | 0 |

*Table 4.* $K_{\max}$ values and count distributions for color-conditioning experiments (Figure 14). Unlike the location-conditioning experiments which use an automated classifier, $K_{\max}$ is evaluated here by *manual* counting over 64 samples, since correct color attribution cannot be reliably assessed by an object-count classifier alone. Columns after the double line represent number of objects generated with correct colors (counts are evaluated at $K = K_{\max}$).

is also conditioned on its absolute location as described in Section E.3. We find that this model struggles with location accuracy even in-distribution (up to 3 objects) and this inaccuracy persists out-of-distribution object counts. Nevertheless the local model of Exp. 3L still displays much stronger length-generalization than the original model of Exp. 3: when trained on 1-3 objects, the original model struggles to ever produce more than 3 objects at any locations, while the local model can draw at least 12 objects (though not always at the desired locations). For Exp. 3 trained on 1-3 objects, we have $K_{\max} = 3$, while for Exp. 3L $K_{\max} = 9$ (note that $K_{\max}$ requires approximately correct locations so it is lower than the maximum number of objects the model can actually produce in this case). $K_{\max}$ values are included in Table 1.

### E.5. Color-conditioning experiments

Table 4 gives $K_{\max}$ values and count distributions for the color-conditioning experiments of Figure 14. Unlike the location-conditioning experiments which use an automated classifier, the color experiment $K_{\max}$ values were evaluated by manually counting over 64 samples per configuration. We use the same $K_{\max}$ definition as in Section E.2 but with a 25% success threshold (generating $K$ objects at least 25% of the time, and at least $K - 2$ objects at least 90% of the time), with objects appearing with approximately correct colors and with acceptable image quality. We test on up to $K = 8$ colors.

## F. Pixel- and Conditional-Locality Gradient Analyses

In this section we describe the pixel- and conditional-locality analyses used in Figures 2, 3, 5, 12, 13.

Pixel-locality is measured as pixel gradient magnitude (average absolute of Jacobian from one pixel to all other pixels), and conditional locality is measured as conditional gradient magnitude (with the "gradient" estimated via a finite difference of the score computed with and without each conditioner). Further detail follows.

**Pixel Gradient Magnitude** We follow Niedoba et al. (2024) in measuring the average absolute of Jacobian from one pixel to all other pixels. The gradient computation uses automatic differentiation to compute the derivative of each output channel at the target pixel with respect to all input pixels. Specifically, for a model output $\widehat{x}_0 = f_\theta(x_t, t, c)$ with shape $[B, C, H, W]$, we compute:

$$G_{i,j}(x,y,t) = \frac{1}{B} \sum_{b=1}^{B} \sum_{c=1}^{C} \left| \frac{\partial \widehat{x}_0[b,c,y,x]}{\partial x_t[b,c,i,j]} \right| \tag{11}$$

where $(x, y)$ is the target pixel and $(i, j)$ ranges over all input pixels. The spatial gradient map $G \in \mathbb{R}^{H \times W}$ indicates how strongly each input location influences the prediction at the target pixel.

**Conditional Gradient Magnitude** To measure the influence of each conditioning label on the prediction at a target pixel location, we approximate the "conditional gradient" via a finite difference by computing output differences when individual labels are ablated from the conditioning set.

Given a conditioning set $C = \{c_1, c_2, \ldots, c_K\}$, we compute the influence of condition $c_k$ at target pixel $(x, y)$ as:

$$I_k(x,y,t) = \| f_\theta(x_t, t, C)[x,y] - f_\theta(x_t, t, C_{-k})[x,y] \|_1 \tag{12}$$

where $C_{-k}$ represents the conditioning set with label $k$ removed, and the norm is taken across color channels. The method is implemented using forward passes only (no actual gradients are required).

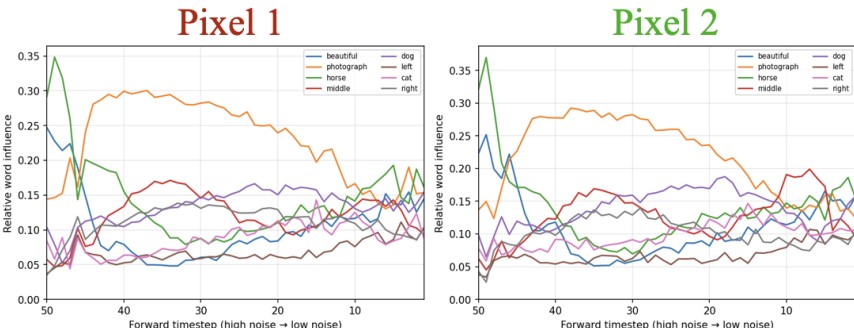

*Figure 15.* **SDXL conditional-locality in pixel-space.** Per-word conditional influence (token-replacement ablation) over denoising timesteps at two pixel locations for the prompt in Figure 5. The nearly identical curves at different pixel locations indicate that conditional-locality is largely absent in pixel-space. Averaged over 10 seeds.

**Locality Metrics** In Figure 13 we directly plot the gradient influence $I_k(x, y)$ for each conditioner $k$ across all times $t$. We quantify spatial locality by finding the smallest square centered at each target location that contains a specified fraction (90%) of the total gradient energy, i.e. the smallest square s.t. $\sum_{(i,j)\in\text{square}} G_{i,j}^2 \geq 0.9 \times \sum_{i,j} G_{i,j}^2$.

In Figure 3, Conditional Locality is an aggregated locality metric obtained by calculating the conditional gradient magnitude at each conditioned location, and computing

$$\text{Conditional locality} = \frac{\sum_k I_k(x_k, y_k, t)}{\sum_k \sum_{k'} I_k(x_{k'}, y_{k'}, t)}.$$

## G. SDXL experiment details.

### G.1. Pixel-space locality

For Figure 5 we use a pretrained SDXL model (out-of-the-box, no finetuning), specifically `stabilityai/stable-diffusion-xl-base-1.0`, with the prompt "a beautiful photograph with a horse in the middle, a dog on the left, and a cat on the right" – which contains implicit location conditioning. We perform a locality analysis similar to the one we used for CLEVR as described in Section F, but with a few adaptations. The gradient computation is performed in the VAE latent space. For conditional influence, we use a token-replacement ablation: for each non-stop-word in the prompt, we replace its token embedding(s) with the padding token embedding while keeping the sequence length fixed, then measure the change in the U-net prediction. This avoids re-encoding a modified prompt string and ensures the only difference is the ablated token. The difference is computed in float32 precision (even though the U-net runs in float16) to preserve small but meaningful spatial variations. All results are averaged over 10 seeds with deterministic noise for reproducibility. We ran the model with 50 inference steps using the standard `EulerDiscreteScheduler`. For each analysis timestep, we add noise to clean latents generated by the full pipeline (with classifier-free guidance) and perform a single-step U-net comparison, computing pixel Jacobians via autograd and word influence via the ablation described above. Results are shown at timesteps $t = 1, 25, 50$ (low, mid, high noise). Additional single-seed results, alternative prompts, and spatial word-influence heatmaps are shown in Section G.1.1.

#### G.1.1. ADDITIONAL PIXEL-SPACE LOCALITY RESULTS

We present additional pixel-space locality results for SDXL across different seeds and prompts. Figure 16 extends the main-text analysis (Figure 5) to additional seeds and prompts. Figure 17 shows spatial word-influence heatmaps at low noise ($t = 1$) and high noise ($t = 50$) for two seeds of the three-object prompt used in the main text, as well as the swapped version (cat and dog positions exchanged). While some weak spatial localization of word influence is occasionally visible in individual heatmaps, this effect is subtle and inconsistent across seeds, prompts, and noise levels. Overall, the quantitative per-word influence profiles over time (Figure 5, right) are nearly identical at different pixel locations, confirming the absence of robust conditional-locality in pixel-space. This motivates the feature-space analysis in Section G.2.

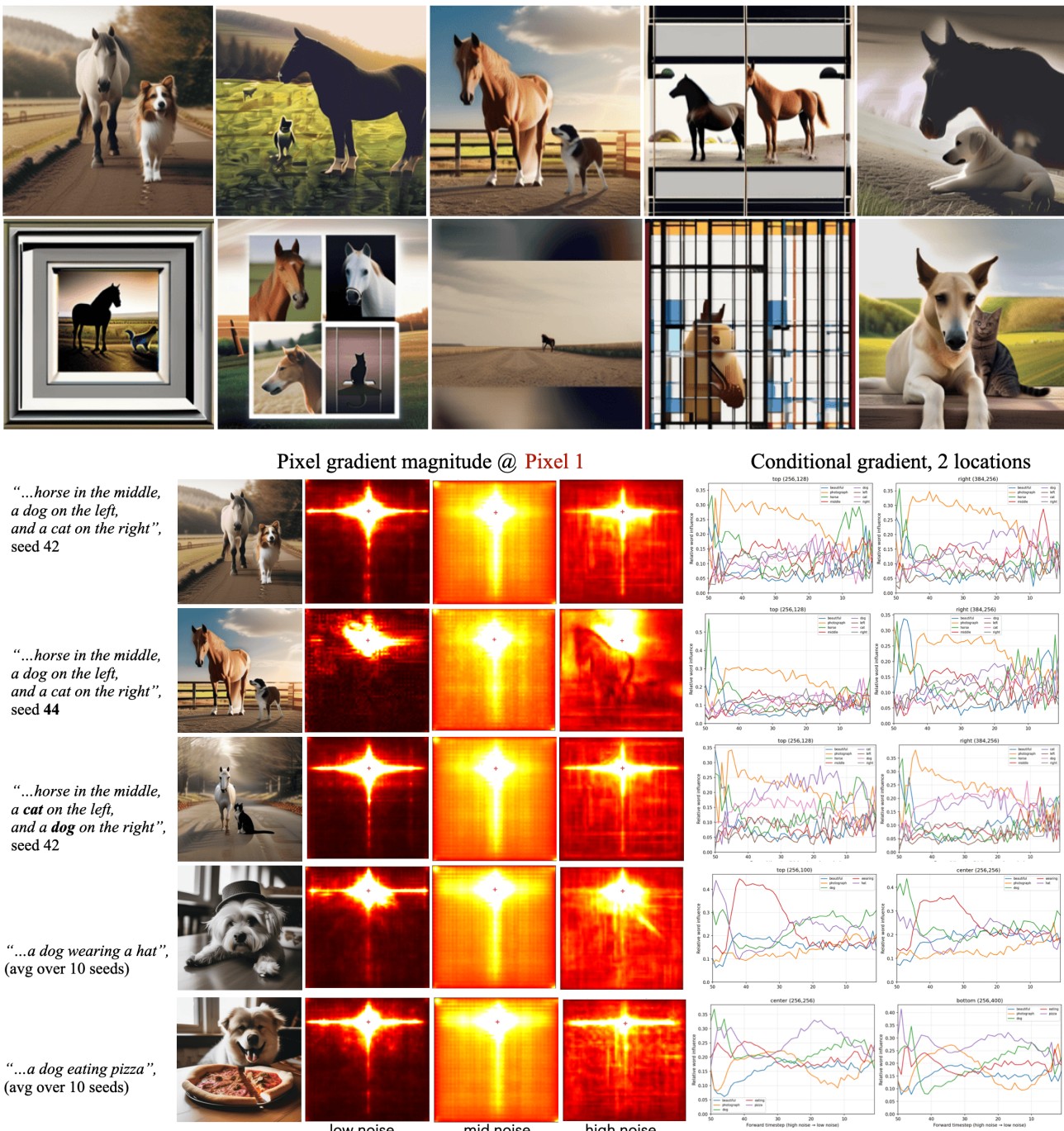

*Figure 16.* **SDXL pixel-space locality across seeds and prompts.** (Top) All 10 seeds used for the main-text analysis (Figure 5) of the prompt "a beautiful photograph with a horse in the middle, a dog on the left, and a cat on the right." (Bottom) Each row shows, for a given prompt and seed: the generated image (left column), pixel gradient magnitude heatmaps at low, mid, and high noise for a single pixel location (center columns, showing pixel-locality via localized cross patterns), and per-word conditional influence (ablation) over time at two pixel locations (right columns). Rows 1–3 are single-seed results for the three-object prompt (two seeds) and its swapped variant; rows 4–5 are 10-seed averages for attribute prompts ("a dog wearing a hat" and "a dog eating pizza") which lack explicit location words. Across all seeds and prompts, the word-influence curves at different pixel locations are nearly identical, confirming the absence of conditional-locality in pixel-space.

**Seed 42: "...horse in the middle, a dog on the left, and a cat on the right"**

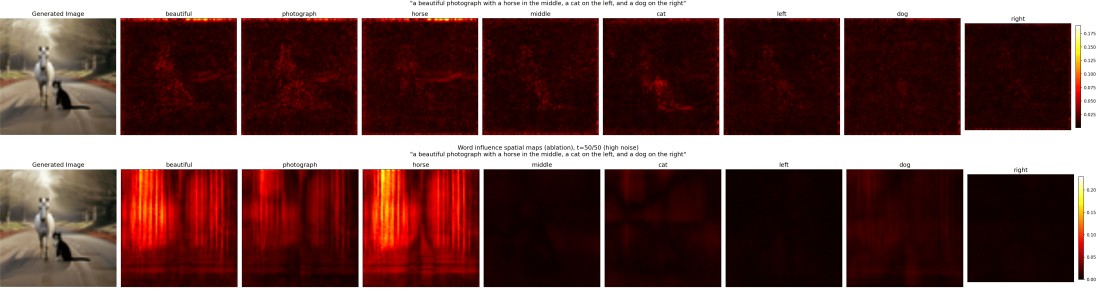

**Seed 44: "...horse in the middle, a dog on the left, and a cat on the right"**

**Seed 42: "...horse in the middle, a cat on the left, and a dog on the right" (swap)**

*Figure 17.* **Spatial word-influence heatmaps in SDXL (single seeds).** For each prompt and seed, word influence (token-replacement ablation) is shown at $t = 1$ (low noise) and $t = 50$ (high noise). Heatmaps use a shared colorscale across words within each timestep. Some weak spatial localization of word influence is occasionally visible, but it is not consistent across seeds, prompts, or noise levels. Seed 42, prompt 0 corresponds to a single seed of the 10-seed average shown in Figure 5.

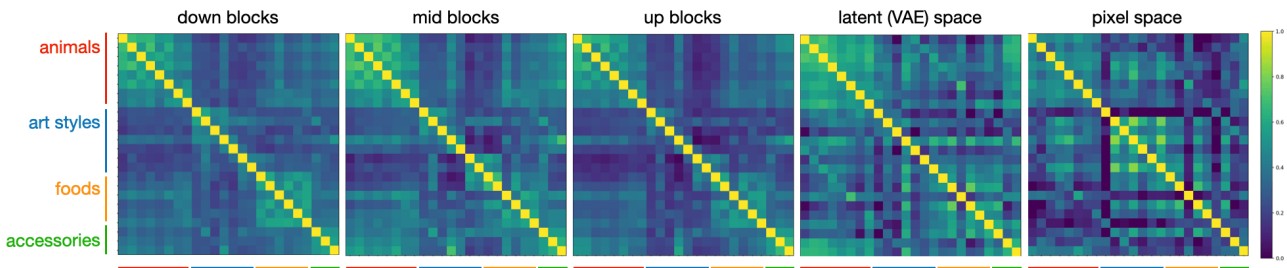

*Figure 18.* Additional detail for Figure 6. $24 \times 24$ cosine similarity heatmaps (F-LCS heuristic, Lemma 3.5) for all 24 prompts across hidden-state activations in the down, mid, and up blocks (at high noise) and output spaces (latent/VAE, pixel). Prompts are grouped by category: animals (dog, cat, horse, rabbit, elephant, owl, fish, butterfly), art styles (monet, vangogh, hokusai, rembrandt, picasso, warhol, pixel_art), foods (sushi, croissant, coffee, pizza, ramen, cake), and accessories (red_hat, sunglasses, scarf). Block-diagonal category structure with crisper boundaries is visible in feature-space (especially down and mid blocks), while output spaces show weaker or absent structure. Within the art styles category, sub-cluster structure is also visible, with impressionist styles (monet, vangogh, hokusai, rembrandt) grouping separately from modern styles (picasso, warhol, pixel_art).

### G.2. Feature-space disentanglement

For Figure 6 we used the SDXL model (`stabilityai/stable-diffusion-xl-base-1.0`) as above. We used the following 24 prompts for the feature-space analysis, organized into 4 semantic categories (with shorthands used in figures):

- **Animals** (8): dog, cat, horse, rabbit, elephant, owl, fish, butterfly – e.g. "A dog, full body, highly detailed photograph."

- **Art styles** (7): monet, vangogh, hokusai, rembrandt, picasso, warhol, pixel_art – e.g. "An oil-painting in the style of Van Gogh."

- **Foods** (6): sushi, croissant, coffee, pizza, ramen, cake – e.g. "Eating sushi with chopsticks, highly detailed photograph."

- **Accessories** (3): red_hat, sunglasses, scarf – e.g. "Wearing a red hat."

- unconditional: "" (empty string baseline).

To measure the F-LCS disentanglement heuristic (Lemma 3.5), we extract hidden-state activations (post-FFN outputs) from each `BasicTransformerBlock` in the SDXL U-net using PyTorch forward hooks. We generate 10 samples per prompt using 15 inference steps with the standard `EulerDiscreteScheduler`. For each prompt and seed, we record hidden states at each timestep and bin them into high ($t > 666$), mid ($333 < t \leq 666$), and low ($t \leq 333$) noise levels. We then average features across seeds, subtract the unconditional baseline, flatten each layer's activations, and compute pairwise cosine similarities between all prompt pairs. We summarize disentanglement via within-category (mean cosine similarity among prompts in the same category) vs. between-category (mean cosine similarity among prompts in different categories) metrics, and their ratio. As baselines, we also compute the same metrics in the VAE latent space (the U-net output latents) and in pixel space (after VAE decoding), to verify that disentanglement is stronger in the network's internal feature-space than in output spaces.

## H. Testing OOD prompts with a model trained on Flickr

Since the train sets of many large-scale text-to-image models like SDXL are not publicly known, the extent to which they are truly capable of OOD generalization is unclear. We therefore study a model for which the training set is known: Gu et al. (2023a)'s Matryoshka Diffusion Model (MDM) trained on subset of 50M Flickr images. In order to study OOD generalization, we designed candidate prompts and searched through the training set captions for keyword matches. We found some evidence of OOD composition on prompts with no conceptually-similar counterpart in the training set. Notably, we found that "a cat eating sushi" actually *does* appear in the train set;[4] however, other similar prompts such as "a dog

---

[4]A train-set caption conceptually matching cat+sushi is: "A room with a wall painted with a mural of a cat eating sushi. The wall has a banner at the top with the words "DIADEMANG" written on it. The room has two low tables with cushions on the floor, and plates of sushi are placed on the tables. The lighting in the room is dim."

eating a croissant" or "a horse eating sushi" do not appear. One OOD example was "a cow jumping over a candlestick": no conceptually similar prompts were found in the train set[5], and yet the model is able to produce some plausible samples.

In Figure 19 we show several OOD example generations from MDM, as well as a feature-space analysis; we compare this to SDXL (although it is not known if the prompts are OOD for SDXL). We used the following prompts: "a cow jumping over a candlestick," "a watercolor painting of a cow jumping over a candlestick," "a dog eating a croissant," "a watercolor painting of a dog eating a croissant," "an oil-painting of an octopus flying through outer space,", "an oil-painting of cat eating sushi with chopsticks." (Only the last prompt is in-distribution for the Flickr training set; for all other compositional prompt a keyword search of the train set found no conceptual matches). MDM is evidently capable of some degree of compositional generalization, generating at least some plausible samples for the OOD prompts. The quality and prompt-fidelity of SDXL is higher, though we do not know whether the prompts are OOD for SDXL.

Figure 19 also shows feature-space cosine similarity experiments. We measure the Lemma 3.5 heuristic on hidden-state activations (post-FFN outputs of each `SelfAttention` module) in the innermost U-net of MDM, using 12 single-concept prompts spanning 6 semantic categories (4 animals, 2 art styles, 2 actions, 1 setting, 1 object, 2 foods), with 10 seeds per prompt and a single forward pass at high noise ($t = 999$). We subtract the unconditional baseline before computing pairwise cosine similarity. The results show clear categorical disentanglement: within-category cosine similarity substantially exceeds between-category similarity across all blocks. For comparison, we run the same prompts through SDXL and find a qualitatively similar pattern at high noise, with even larger within/between gaps. These experiments offer preliminary – though far from conclusive – evidence that OOD compositional generalization is actually possible, and that compositional structure in feature-space may support it.

---

[5]Two train-set captions matching the keywords "cow" and "candlestick" were found, for example: "A page from an old book with various crests, including one with scissors and a scissor-like symbol, one with a cow, one with a candlestick, one with a statue of a man holding a heart, and one with a shield with a cross and the words 'La Comte Des Marechaux.'..." but neither conceptually represented a cow jumping over a candlestick.

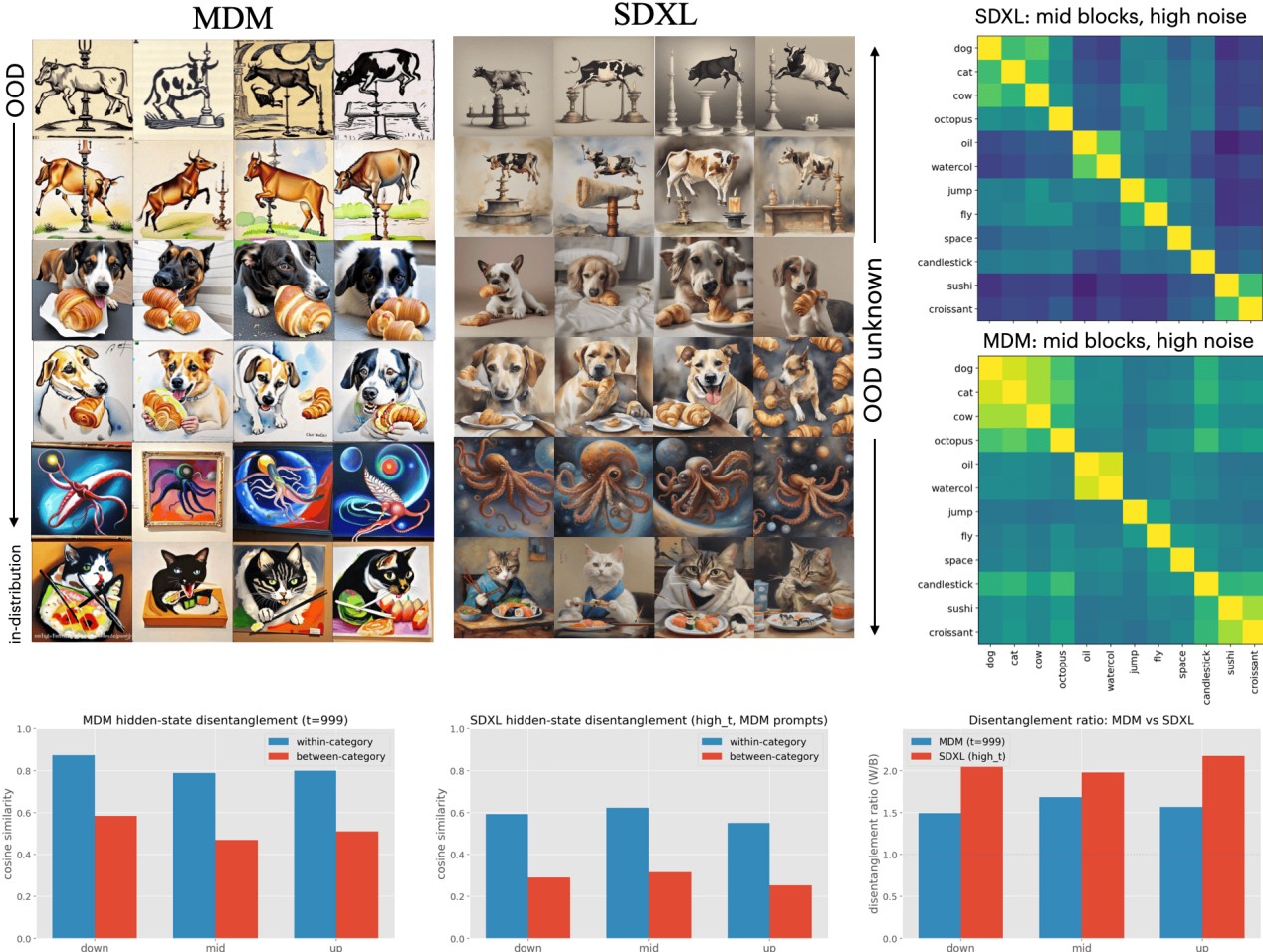

*Figure 19.* **MDM generalization on OOD prompts.** (Top) Sample generations from MDM (left) and SDXL (right) for compositional prompts. Rows are ordered from OOD (top) to in-distribution (bottom) for the MDM Flickr training set; OOD status for SDXL is unknown. (Middle right) Cosine similarity heatmaps of hidden-state activations (mid blocks, high noise) for SDXL and MDM, showing block-diagonal category structure (animals, styles, actions, foods). (Bottom) Within- vs. between-category cosine similarity (F-LCS heuristic, Lemma 3.5) across down, mid, and up blocks for both MDM ($t = 999$) and SDXL (high noise), and a comparison of their disentanglement ratios. Both models show clear categorical disentanglement, with within-category similarity substantially exceeding between-category similarity across all blocks.

