# OpenReview forum: "Local Mechanisms of Compositional Generalization in Conditional Diffusion"
_ICML.cc/2026/Conference — ICML 2026 spotlight_

### Official Review · Reviewer_JDqx · 2026-03-08

**Soundness:** 3
**Presentation:** 3
**Significance:** 3
**Originality:** 3
**Overall Recommendation:** 5
**Confidence:** 3

**Summary:**

This paper studies the mechanism behind composiional generalization of diffusion models. The authors point out that the equivalence between conditional projective composition and local conditional scores can explain compositional generalization. Further, the authors use location-conditioned generation, causal intervention, color-conditioned generation, and real-world SDXL model as supporting evidence to validate the theoretical contribution.

**Compliance With Llm Reviewing Policy:**

Affirmed.

**Final Justification:**

With the improvement to the manuscript, I believe this paper will help many fellow researchers.

**Key Questions For Authors:**

1. I am curious why SDXL is picked to provide the evidence of the real-world models.
2. Following question 1, could the authors provide some discussion on what might be observed from other real-world models, considering architectural difference between U-Net and DiT or training difference between DDPM and flow matching?

**Limitations:**

yes

**Strengths And Weaknesses:**

Pros:
1. The presentation feels quite good. While maintaining theoretical rigor, the authors also make the theoretical part easy to understand. I especially like how the authors use figure 4 to make readers understand the definitions.
2. The proposed theoretical framework is supported by extensive evidence, covering both correlation and causality, and even extending to real-world models.

Cons:
1. Section 3.1 could use some further discussion or interpretation of the equivalence to draw a clearer and more direct conclusion. Currently, it feels a bit like: I know there is the equivalence, but what does that imply? Similarly, a direct interpretation and summary of the final conclusion (why diffusion models can compositionally generalize) in the introduction section would help.
2. There is some repeated content, such as very long captions and main content stating the same thing, or the starting paragraph of section 3. By removing such repeated content, some frequently referenced content in the Appendix might be able to be moved to the main content, such as figure 8.
3. The fonts in certain figures, such as figure 6, are too small.

---

> ### Author Rebuttal · Authors · 2026-03-29
>
> Thank you for the thoughtful review and constructive feedback. We
> respond to each of your questions and suggestions below.
>
> **Concern 1: "Section 3.1 could use some further discussion or interpretation of the equivalence to draw a clearer and more direct conclusion."**
>
> We appreciate this feedback. We will add a new paragraph
> after the CPC-LCS equivalence (Lemma 3.3) that clarifies the key
> implications:
>
> > "This equivalence has two important implications. First, it means
> > that locality in the score function is both necessary and sufficient
> > for compositional factorization: if a trained model exhibits local
> > conditional scores, it has implicitly learned a compositional
> > structure, and vice versa. Second, beyond this equivalence, we are interested
> > in whether LCS could be a causal mechanism for compositional
> > generalization. The CPC⇔LCS equivalence is necessary but not
> > sufficient to establish causality; the causal intervention
> > experiments in Section 4 provide direct evidence that enforcing LCS
> > can cause compositional generalization."
>
> We will also added a concise summary in the introduction:
>
> > "This equivalence means that locality in the score function is both
> > necessary and sufficient for compositional factorization — observing
> > one implies the other — and suggests that LCS may be a causal
> > mechanism for compositional generalization, which we test via direct
> > intervention."
>
> **Concern 2: "There is some repeated content, such as very long captions and main content stating the same thing, or the starting paragraph of section 3. By removing such repeated content, some frequently referenced content in the Appendix might be able to be moved to the main content, such as figure 8."**
>
> We will reduce the redundancy for the final draft, and use the extra space to expand the
> SDXL and MDM experiments with more rigorous quantitative analysis.
>
> **Concern 3: "The fonts in certain figures, such as figure 6, are too small."**
>
> Thank you, we will fix in final version.
>
> **Question 1: "I am curious why SDXL is picked to provide the evidence of the real-world models."**
>
> SDXL was chosen as a widely-used, well-studied open-source model that
> enables reproducible analysis. We have also validated our findings on
> the Matryoshka Diffusion Model (MDM) trained on Flickr, which uses a
> different architecture and a known training set, finding qualitatively
> similar feature-space disentanglement structure.
>
> **Question 2: "Could the authors provide some discussion on what might be observed from other real-world models, considering architectural difference between U-Net and DiT or training difference between DDPM and flow matching?"**
>
> - The equivalence between CPC/LCS is
> architecture-agnostic. Whether locality is learned may be somewhat architecture-dependent
> -- e.g. Kamb & Ganguli (2024) argue that locality is strongly tied to convolutional (U-Net) architectures --
> but may also be driven by the underlying structure of the data.
> Niedoba et al. (2024) perform experiments on a variety of
> architectures (including DiT and others) and find that locality holds
> across all of them, suggesting the phenomenon is not U-Net-specific.
> Lukoianov et al. (2025) further argue that locality arises from
> statistical properties of the data rather than network inductive bias.
> Given the evidence presented by Niedoba and Lukoianov, we expect that
> DiT-based architectures would likely exhibit some degree of compositional structure
> in feature-space (L-FCS) similar to what we observed for SDXL and MDM (U-Nets),
> although it would certainly be interesting to confirm whether U-Net architectures
> confer any advantage over DiTs in learning local structure.
>
> - The velocity field learned by flow matching is closely related to
>   the score function (differing by a known transformation depending on
>   the noise schedule), so the CPC/LCS theory applies analogously to
>   flow matching models. We would expect to observe similar locality
>   properties for models trained via flow matching;
>   confirming this and noting any differences would be a natural direction for future
>   work.

---

> > ### Author Rebuttal · Reviewer_JDqx · 2026-04-03
> >
> > With the improvement to the manuscript, I believe this paper will help many fellow researchers

---

### Official Review · Reviewer_jCxF · 2026-03-11

**Soundness:** 4
**Presentation:** 2
**Significance:** 4
**Originality:** 4
**Overall Recommendation:** 6
**Confidence:** 3

**Summary:**

This work studies compositional (length) generalization in image diffusion models, finding in a controlled toy setting that generalization _sometimes_ works. The authors show theoretically that locality (sparse dependence on pixels and arbitrary conditioners) is a key mechanism for compositional generalization. The authors provide ample empirical evidence, from generalization experiments on CLEVR, to a causal study, to structural analysis of pretrained models, to support their findings.

**Compliance With Llm Reviewing Policy:**

Affirmed.

**Final Justification:**

My (minor) concerns have been addressed; I recommend acceptance of this interesting and relevant work.

**Key Questions For Authors:**

1. The sparse dependency condition reminds me of a series of works from Brady et al. [1, 2, 3], which also partially deal with compositional generalization. In these works, sparse dependencies are usually expressed via structured (block-diagonal) Jacobians. Do you see a connection between some of these approaches and results and this work?

[1] Provably Learning Object-Centric Representations; J Brady*, RS Zimmermann*, Y Sharma, B Schölkopf, J von Kügelgen, ...; ICML 2023 (Oral)

[2] Provable Compositional Generalization for Object-Centric Learning; T Wiedemer*, J Brady*, A Panfilov*, A Juhos*, M Bethge, W Brendel; ICLR 2024 (Oral)

[3] Interaction Asymmetry: A General Principle for Learning Composable Abstractions; J Brady, J von Kügelgen, S Lachapelle, S Buchholz, T Kipf, W Brendel; ICLR 2025

**Limitations:**

The limitations of this work are clearly and adequately outlined in the manuscript.

**Strengths And Weaknesses:**

# Strengths
1. This paper addresses important questions in image generation (why does compositionality sometimes work? how can we build models that generalize provably?) in an interesting and convincing way.
2. To the best of my knowledge, the proposed theoretical framework and the established connection between prior works is novel.
3. The presented theoretical contribution seems sound.
4. The authors provide sufficient evidence in controlled settings and via different approaches to validate and explain their theory. I specifically appreciate the causal intervention study in addition to the initial CLEVR experiments to directly establish local conditional scores as a "knob to turn" in model development to improve generalization.
5. While still limited in scope, the analysis of pretrained models that reveals compositional structures in pixel- and feature-space attest to the applicability of the theory to current diffusion model research.

# Minor weaknesses
1. It is not entirely clear to me why the authors focus mainly on length generalization, rather than other types of compositions (such as location-shape, shape-color, etc.).
2. I find many of the schematic figures hard to interpret (e.g., Fig. 4). A clearer explanation of what is illustrated here could improve the accessibility of this paper.
3. The paper is written at a high technical level, but as a result sometimes very dense and hard to follow. Breaking up longer paragraphs and providing additional intuition to some of the theory might improve the accessibility of the paper.
4. The figure order doesn't match the order of points in the text, forcing the reader to jump back-and-forth a lot.

Overall, I find this work to be a very strong and meaningful contribution to the field and recommend acceptance. However, additional work could be done to better explain the theory and make the work more accessible to a general audience.

---

> ### Author Rebuttal · Authors · 2026-03-29
>
> Thank you for your support for this paper and the thoughtful,
> actionable feedback. We respond to each of your points below.
>
> **Minor Weakness 1: "It is not entirely clear to me why the authors focus mainly on length generalization, rather than other types of compositions (such as location-shape, shape-color, etc.)."**
>
> Length generalization is intended to be a controlled, verifiable
> special case of compositional generalization (the broader topic of the
> paper). A particular advantage of the CLEVR location
> length-generalization setting is that the compositional structure
> holds in pixel-space, making it possible to rigorously study both the
> theory (CPC/LCS in pixel-space) and the experiments (directly
> measurable locality). Compositions such as location-shape,
> shape-color, or style+content are examples of compositions between
> concepts that typically require compositional structure in
> feature-space rather than pixel-space. Our SDXL and MDM experiments
> are intended to explore this kind of feature-space composition.
>
> **Minor Weakness 2: "I find many of the schematic figures hard to interpret (e.g., Fig. 4). A clearer explanation of what is illustrated here could improve the accessibility of this paper."**
>
> We will revise the Figure 4 caption to be clearer as follows:
>
> > "Conditional projective composition (CPC) and local conditional
> > scores (LCS), illustrated for an image with four conditions c_0,
> > c_1, c_2, c_3 (e.g. four object locations). A CPC factorizes the
> > conditional distribution independently over disjoint pixel subsets
> > M_0, M_1, M_2, M_3, M_b: the pixels in M_j depend only on
> > condition c_j, and the 'background' pixels M_b are unconditional.
> > LCS is the corresponding property of the score function: the score
> > at each pixel i depends only on a subset N_i of nearby pixels and a
> > subset L_i of relevant conditions. The CPC⇔LCS equivalence (Lemma
> > 3.3) shows that these two perspectives — compositional structure in
> > the distribution and locality in the score — are equivalent."
>
>
> **Minor Weaknesses : "... providing additional intuition to some of the theory might improve the accessibility of the paper."**
>
> We will add an interpretive paragraph after the CPC-LCS
> equivalence (Lemma 3.3) that spells out the two key implications: (1)
> locality in the score is both necessary and sufficient for
> compositional factorization, so observing one implies the other, and
> (2) we are interested in whether LCS could be a causal mechanism for
> compositional generalization — the equivalence is necessary but not
> sufficient for causality, while the causal intervention experiments
> provide direct evidence. The new text is given in the response to Reviewer JDqx.
>
> **Question: "The sparse dependency condition reminds me of a series of works from Brady et al. [1, 2, 3], which also partially deal with compositional generalization. In these works, sparse dependencies are usually expressed via structured (block-diagonal) Jacobians. Do you see a connection between some of these approaches and results and this work?"**
>
> We thank the reviewer for making this excellent connection. The Brady
> et al. line of work identifies closely related structural conditions
> for compositional generalization in the autoencoder setting (in
> contrast to our setting of conditional diffusion models). Their
> block-diagonal Jacobian condition [2] is analogous to our LCS, their
> slot-based generative models [1] correspond to our CPC, and their
> disentangled latent representations [1, 2] correspond to our
> feature-space F-CPC/F-LCS. Their results in [2] implicitly leverage a
> CPC-LCS-type equivalence; our work makes this explicit and proves it
> for conditional score functions. We have added citations and a
> discussion of this connection. We will add the following to the theory section:
>
> > "We note that a related CPC⇔LCS-type connection has been implicitly
> > leveraged in the object-centric representation learning literature,
> > where block-diagonal Jacobians (analogous to LCS) and slot-based
> > generative models (analogous to CPC) have been shown to support
> > compositional generalization in autoencoders (Brady et al., 2023;
> > Wiedemer et al., 2024; Brady et al., 2025); our work makes this
> > equivalence explicit and proves it in the conditional score function
> > setting."

---

> > ### Author Rebuttal · Reviewer_jCxF · 2026-04-06
> >
> > I thank the authors for the insightful rebuttal. I only had minor concerns and already recommended acceptance, but have further raised my score as I believe this is an interesting work that is of great interest to the community.

---

### Official Review · Reviewer_ygd5 · 2026-03-12

**Soundness:** 3
**Presentation:** 4
**Significance:** 3
**Originality:** 3
**Overall Recommendation:** 4
**Confidence:** 3

**Summary:**

The authors introduce a controlled setting to study compositional generalization via length generalization in conditional diffusion models. By extending prior work on local score functions (Kamb & Ganguli; Niedoba et al.) to the conditional setting, they define Local Conditional Scores (LCS). The paper connects two lines of work: local scores and projective composition (Bradley et al., 2025), by introducing Conditional Projective Composition (CPC) for a single conditional distribution. The authors theoretically show an equivalence between CPC and LCS, linking compositional structure with locality in score functions. They empirically study these ideas using controlled CLEVR-based experiments on conditional diffusion models.

**Compliance With Llm Reviewing Policy:**

Affirmed.

**Key Questions For Authors:**

The results are demonstrated mainly in controlled CLEVR settings. Could the authors comment on how far the proposed mechanism is expected to generalize to more realistic datasets and large-scale diffusion models? If additional empirical evidence is available, including it would strengthen the paper.

**Limitations:**

yes

**Strengths And Weaknesses:**

##Strengths

- **Clear logic.**

  By introducing Conditional Projective Composition (CPC) and Local Conditional Scores (LCS) for conditional diffusion models, the authors provide a unified framework linking compositional structure and score locality. While the contribution is not very large in scope, it is a clear  theoretical step.

- **Simple experimental setup.**

  Using a minimal setup where the number of objects differs between training and testing provides a clean environment to analyze compositional behavior. The intervention experiment based on this setup (e.g., Fig. 1) offers useful insights.
- The paper is clearly written and easy to follow.

---

## Weaknesses

- **Incremental theoretical novelty.**

  The idea that locality may support compositional generalization has appeared in several prior works. While the paper provides a new theoretical perspective on this connection, the core intuition is not entirely new.

- **Experimental setup builds on existing benchmarks.**

  The experiments rely on the CLEVR dataset and length generalization, both of which have been used in prior work. Although the paper frames them in a new context, the empirical setup itself is standard.

- **Strong assumptions and unclear generality.**

  The theory relies on several structural assumptions, and it is unclear how well the conclusions generalize to more realistic settings. The authors suggest that pixel-space locality may extend to feature-space locality, but the evidence for this claim remains limited. The feature-space compositionality analysis and SDXL experiments are interesting but still exploratory.

---

> ### Author Rebuttal · Authors · 2026-03-29
>
> Thank you for the supportive review and helpful feedback. We respond
> to each of your concerns below.
>
> **Weakness 1: "Incremental theoretical novelty. The idea that locality may support compositional generalization has appeared in several prior works. While the paper provides a new theoretical perspective on this connection, the core intuition is not entirely new."**
>
> While the broad intuition that
> locality may support generalization has appeared before, our specific
> contributions are new:
>
> 1. The formal **CPC-LCS equivalence theorem** (Lemma 3.3) is, to our
> knowledge, the first proof that compositional factorization and local
> conditional scores are equivalent — not just correlated or
> heuristically connected.
> 2. The **causal intervention experiments** (Exp. 2L, 3L) provide the
> first direct evidence that enforcing locality may *cause*
> compositional generalization.
> 3. The **feature-space extension** (F-CPC/F-LCS) precisely connects
> compositional structure with disentanglement.
>
> **Weakness 2: "Experimental setup builds on existing benchmarks. The experiments rely on the CLEVR dataset and length generalization, both of which have been used in prior work. Although the paper frames them in a new context, the empirical setup itself is standard."**
>
> CLEVR and length generalization are indeed established
> benchmarks, but provide a controlled setting enabling rigorous causal analysis. The novelty of our experiments lies in the direct measurement of LCS structure in
> trained models and the causal intervention experiments that test locality
> as a mechanism of compositional generalization.
>
> **Weakness 3: "Strong assumptions and unclear generality. The theory relies on several structural assumptions, and it is unclear how well the conclusions generalize to more realistic settings. The authors suggest that pixel-space locality may extend to feature-space locality, but the evidence for this claim remains limited."**
>
> We note that the theoretical assumptions can be relaxed somewhat:
> Lemma C.2 in Appendix C shows that the score of an *approximately*
> CPC distribution is *approximately* an LCS, with the approximation
> improving at higher noise. This means perfect disjoint pixel subsets
> are not required — the theory applies in an approximate sense to more
> realistic distributions.
>
> Regarding generality to real-world models, we have substantially
> strengthened the SDXL and MDM experiments:
>
> - **SDXL feature-space**:
>   - *Methodology*: We now analyze post-FFN hidden-state activations
>     (rather than cross-attention maps used in the original
>     submission), providing a more direct probe of the network's
>     internal representations.
>   - *Scale*: 24 prompts spanning 4 semantic categories with 10 seeds
>     (compared to a smaller set in the original submission).
>   - *Quantitative analysis*: Within-category cosine similarities
>     substantially exceed between-category similarities
>     (disentanglement ratios of 1.4–1.6x), with an arch-shaped
>     per-layer profile peaking near the U-net bottleneck.
>
>   **SDXL Hidden-State Disentanglement** (24 prompts, 4 categories, 10 seeds):
>
>   | Space | Within-cat cos sim | Between-cat cos sim | Ratio |
>   |---|---|---|---|
>   | Hidden states — down blocks (high noise) | 0.472 | 0.302 | 1.56 |
>   | Hidden states — mid block (high noise) | 0.474 | 0.333 | 1.42 |
>   | Hidden states — up blocks (high noise) | 0.413 | 0.273 | 1.51 |
>   | Latent space (VAE) | 0.464 | 0.353 | 1.31 |
>   | Pixel space | 0.378 | 0.266 | 1.42 |
>
>   Feature-space hidden states show disentanglement ratios of 1.4–1.6x,
>   exceeding both pixel-space and VAE/latent-space baselines.
>
> - **MDM validation**:
>   - *Methodology*: The same hidden-state analysis on a Matryoshka
>     Diffusion Model trained on the known Flickr dataset.
>   - *Quantitative analysis*: Qualitatively similar disentanglement
>     ratios of 1.49–1.68x, confirming the findings generalize beyond
>     SDXL.
>
>   **MDM Hidden-State Disentanglement** (12 prompts, 6 categories, 10 seeds, t=999):
>
>   | Block group | Within-cat cos sim | Between-cat cos sim | Ratio |
>   |---|---|---|---|
>   | down_blocks | 0.873 | 0.584 | 1.49 |
>   | mid_blocks | 0.789 | 0.469 | 1.68 |
>   | up_blocks | 0.800 | 0.511 | 1.57 |
>   | all (aggregated) | 0.817 | 0.526 | 1.55 |
>
> **Question: "The results are demonstrated mainly in controlled CLEVR settings. Could the authors comment on how far the proposed mechanism is expected to generalize to more realistic datasets and large-scale diffusion models? If additional empirical evidence is available, including it would strengthen the paper."**
>
> As discussed above, we have added more robust
> and comprehensive SDXL and MDM experiments. We also provide
> additional pixel-space analysis; please see our response to Reviewer 3Qzn.

---

> > ### Author Rebuttal · Reviewer_ygd5 · 2026-04-01
> >
> > The additional experiments (SDXL/MDM) are valuable and strengthen the empirical support. However, the central concern regarding generalization to realistic settings remains only partially addressed, as the conditions under which the proposed mechanism holds in real-world scenarios are still not clearly characterized. Therefore, I keep my score unchanged.

---

### Official Review · Reviewer_3Qzn · 2026-03-14

**Soundness:** 3
**Presentation:** 3
**Significance:** 3
**Originality:** 3
**Overall Recommendation:** 4
**Confidence:** 3

**Summary:**

This paper investigates length generalization in conditional diffusion models. This paper demonstrates the equivalence between conditional projective composition and local conditional scores. Empirically, the authors show that models capable of length-generalizing exhibit local conditional scores, whereas failing models do not. They also perform a causal intervention to enforce locality, which restores length generalization.

**Compliance With Llm Reviewing Policy:**

Affirmed.

**Final Justification:**

I think the paper is a solid contribution to the community.

**Key Questions For Authors:**

I'd appreciate the authors' thoughts on the weaknesses listed above.

**Limitations:**

Yes

**Strengths And Weaknesses:**

Strengths:

1. The theoretical proofs linking CPC to LCS provide a solid mathematical foundation for understanding compositional generalization.
2. Extending the concept of local scores to flexible conditional distributions, and defining feature-space CPC (F-CPC), is a valuable theoretical contribution to the diffusion literature.

Weaknesses:

1. The empirical validation is overly reliant on the highly controlled, synthetic CLEVR dataset. The evidence provided for SDXL is preliminary and lacks the rigor needed to prove that these theoretical mechanisms strictly hold or can be manipulated in complex, real-world text-to-image generation.

2. Similarly, while the theory is sound, the practical implications for training better large-scale diffusion models are limited. The causal intervention works on toy architectures and scaling this specific architectural constraint to billions of parameters is unexplored.

---

> ### Author Rebuttal · Authors · 2026-03-29
>
> **Weakness 1: “The evidence provided for SDXL is preliminary and lacks the rigor needed to prove that these theoretical mechanisms strictly hold or can be manipulated in complex, real-world text-to-image generation.”**
>
> We have strengthened the SDXL and MDM experiments as follows:
>
> - **SDXL feature-space analysis**:
>   - *Methodology*: We now analyze post-FFN hidden-state activations
>     (rather than cross-attention maps) from each transformer block,
>     providing a more direct probe of the internal
>     representations.
>   - *Scale*: 24 prompts spanning 4 semantic categories (8 animals, 7
>     art styles, 6 foods, 3 accessories) with 10 seeds per prompt
>     (compared to 8 prompts, 1 seed in original submission).
>   - *Quantitative analysis*: Within-category cosine similarities
>     substantially exceed between-category similarities, yielding
>     disentanglement ratios of 1.4–1.6x. Feature-space is more
>     disentangled than both pixel-space and VAE/latent-space. The
>     per-layer disentanglement profile peaks near the U-net bottleneck.
>
>   **SDXL Hidden-State Disentanglement** (24 prompts, 4 categories, 10 seeds):
>
>   | Space | Within-cat cos sim | Between-cat cos sim | Ratio |
>   |---|---|---|---|
>   | Hidden states — down blocks (high noise) | 0.472 | 0.302 | 1.56 |
>   | Hidden states — mid block (high noise) | 0.474 | 0.333 | 1.42 |
>   | Hidden states — up blocks (high noise) | 0.413 | 0.273 | 1.51 |
>   | Latent space (VAE) | 0.464 | 0.353 | 1.31 |
>   | Pixel space | 0.378 | 0.266 | 1.42 |
>
> - **MDM validation**:
>   - *Methodology*: We ran the same hidden-state disentanglement
>     analysis on the Matryoshka Diffusion Model (trained on the known
>     Flickr dataset), which provides a controlled setting where OOD
>     compositions can be verified.
>   - *Quantitative analysis*: Qualitatively similar F-LCS structure
>     with disentanglement ratios of 1.49–1.68x, confirming that the
>     findings are not specific to SDXL.
>
>   **MDM Hidden-State Disentanglement** (12 prompts, 6 categories, 10 seeds, t=999):
>
>   | Block group | Within-cat cos sim | Between-cat cos sim | Ratio |
>   |---|---|---|---|
>   | down_blocks | 0.873 | 0.584 | 1.49 |
>   | mid_blocks | 0.789 | 0.469 | 1.68 |
>   | up_blocks | 0.800 | 0.511 | 1.57 |
>   | all (aggregated) | 0.817 | 0.526 | 1.55 |
>
> - **SDXL pixel-space analysis**:
>   - *Methodology*: We now use a more rigorous token-replacement ablation instead of
>     prompt-string removal (avoids re-encoding prompt).
>   - *Scale*: Averaged over 10 seeds across multiple prompts (original version used a single seed).
>   - *Findings*: While pixel-locality is present, conditional-locality is largely absent in
>     pixel-space — per-word influence profiles are nearly identical at
>     different spatial locations — motivating the feature-space
>     analysis.
>
>   **Pixel-space conditional influence** (3-animal prompt, 10-seed avg).
>   Mean ± max deviation of relative word influence across 4 pixel
>   locations:
>
>   | t | beautiful | photograph | horse | middle | dog | left | cat | right |
>   |---|---|---|---|---|---|---|---|---|
>   | 1 | .150±.009 | .147±.017 | .157±.004 | .094±.002 | .154±.010 | .103±.008 | .096±.012 | .099±.008 |
>   | 25 | .085±.005 | .264±.008 | .100±.001 | .115±.007 | .160±.013 | .052±.004 | .104±.007 | .119±.004 |
>   | 50 | .248±.020 | .151±.015 | .300±.033 | .055±.009 | .090±.020 | .041±.003 | .081±.010 | .040±.004 |
>
>   The max deviation across pixel locations is small (0.002–0.033),
>   confirming the absence of conditional-locality in pixel-space. We
>   also tested swapping "left"/"right" in the prompt and found that it
>   does not swap which pixel location is more influenced by which
>   animal — the spatial-position words have negligible differential
>   effect. Additional prompts ("dog wearing hat", "dog eating pizza")
>   show the same pattern. This motivates the feature-space analysis.
>
> While these experiments do not prove that the theoretical mechanisms
> can be directly manipulated in large-scale models, they provide
> substantially stronger quantitative evidence connecting our theory
> with real-world compositional structure.
>
>
> **Weakness 2: “Similarly, while the theory is sound, the practical implications for training better large-scale diffusion models are limited. The causal intervention works on toy architectures and scaling this specific architectural constraint to billions of parameters is unexplored.”**
>
> We agree that scaling the causal intervention to large-scale models is an
> important direction for future work. Several existing methods can be viewed as implicit
> “local interventions” consistent with our theory: layout-to-image methods that
> use explicit local constraints or biases, or sparse attention
> architectures (Child et al., 2019; Sun et al., 2022). Our theory helps to
> explain why these approaches are beneficial, and suggests targeting
> compositional generalization by specifically enforcing local conditional scores.

---

> > ### Author Rebuttal · Reviewer_3Qzn · 2026-04-03
> >
> > I thank for the authors for the informative response and would like to maintain my positive rating.

---

### Decision · Program_Chairs · 2026-04-30

**Decision:**

Accept (spotlight)

**Comment:**

**Summary and Decision**
While reviewers noted minor concerns regarding empirical validation and theoretic generality, the general consensus is that the paper makes a significant contribution to the theoretic understanding of compositional generalization.

The reviewers specifically highlighted the following strengths:
* Theoretic proofs linking CPC and LCS to provide a solid foundation for compositional generalization.
* Experimental evidence for the developed theory using SDXL and MDM.

**Rebuttal and Discussion**
During the discussion phase, the authors addressed concerns about the empirical validity with new experimental results on SDXL and MDM, which satisfied the reviewers. Given the impact of the work, the Area Chair recommends acceptance.

**Final Instructions to Authors**
Please ensure that the promised revisions from the rebuttal are incorporated into the camera-ready version.